# Interacting with volatile environments stabilizes hidden-state inference and its brain signatures

Aurélien Weiss [1,2,3 ✉], Valérian Chambon [2,4], Junseok K. Lee [1,2], Jan Drugowitsch[5] & Valentin Wyart [1,2 ✉]

Making accurate decisions in uncertain environments requires identifying the generative cause of sensory cues, but also the expected outcomes of possible actions. Although both cognitive processes can be formalized as Bayesian inference, they are commonly studied using different experimental frameworks, making their formal comparison difficult. Here, by framing a reversal learning task either as cue-based or outcome-based inference, we found that humans perceive the same volatile environment as more stable when inferring its hidden state by interaction with uncertain outcomes than by observation of equally uncertain cues. Multivariate patterns of magnetoencephalographic (MEG) activity reflected this behavioral difference in the neural interaction between inferred beliefs and incoming evidence, an effect originating from associative regions in the temporal lobe. Together, these findings indicate that the degree of control over the sampling of volatile environments shapes human learning and decision-making under uncertainty.

[1] Laboratoire de Neurosciences Cognitives et Computationnelles, Institut National de la Santé et de la Recherche Médicale (Inserm), Paris, France. [2] Département d'Études Cognitives, École Normale Supérieure, Université PSL, Paris, France. [3] Université de Paris, Paris, France. [4] Institut Jean Nicod, Centre National de la Recherche Scientifique (CNRS), Paris, France. [5] Department of Neurobiology, Harvard Medical School, Boston, MA, USA. ✉email: aurelienweiss@gmail.com; valentin.wyart@ens.fr

Making accurate decisions in an uncertain environment requires inferring its properties from imperfect information[1,2]. When categorizing an ambiguous stimulus, this "hidden-state inference" process consists in identifying the generative cause of sensory cues. By contrast, when foraging rewards, inference concerns the expected outcomes of possible courses of action. A constitutive, yet rarely considered difference between these two forms of inference lies in the degree of control over information sampling conferred to the decision-maker. Indeed, cue-based inference relies on the presentation of information to an observer interpreting a relevant property of its environment (here, the category of the presented stimulus), whereas outcome-based inference relies on the active sampling of information by an agent interacting with its environment to achieve a particular goal (here, maximizing rewards).

Thus, it remains unclear whether humans learn and decide differently based on the same uncertain information when the information in question corresponds either to external cues or to outcomes of a previous decision. A formal comparison between cue-based and outcome-based inference requires a shared computational framework that describes them. Yet, canonical "sequential-sampling" models of sensory evidence accumulation are cast in terms of a continuous random walk process spanning hundreds of milliseconds[3,4], whereas "reinforcement learning" models of action valuation rely on discrete updates of expected outcomes over much longer timescales[5,6]. Another challenge for a direct comparison between the two types of inference comes from the large differences in the experimental paradigms developed to study perceptual (cue-based) decisions and reward-guided (outcome-based) decisions. In particular, perceptual tasks increase uncertainty by decreasing the signal-to-noise ratio of presented stimuli—e.g., the motion coherence of random-dot kinematograms[7,8]. By contrast, reward-guided tasks increase uncertainty by decreasing the predictability of action-outcome contingencies—e.g., differences in reward probability associated with possible actions[9,10]. Finally, a recently developed paradigm that compares active and passive sampling confers intrinsic benefits to active sampling through improved information gathering[11,12], thereby rendering comparisons between cue-based and outcome-based inference difficult.

To overcome these challenges, we designed and tested an adaptive decision-making task based on reversal learning[13,14] and a computational framework based on Bayesian inference[15–17], in which cue-based and outcome-based inference can be framed and compared in tightly matched conditions. We recorded magnetoencephalographic (MEG) signals to identify the neural representations and dynamics supporting the two types of inference and their differences. We obtained converging behavioral and neural evidence that interacting with uncertain information stabilizes hidden-state inference, as if humans perceive volatile environments as more stable when interacting with uncertain outcomes than when observing equally uncertain cues.

## Results

**Reversal learning task**. Healthy adult participants ($N = 24$) performed a reversal learning task based on visual stimuli, which we framed either as cue-based or outcome-based inference, in two conditions corresponding to different blocks of trials. In both conditions, participants were asked to track a hidden state of the task, which alternates occasionally and unpredictably between two discrete values (Fig. 1). Stimuli corresponded to oriented bars drawn from one of two overlapping probability distributions (categories) centered on orthogonal orientations, each associated with a color (Fig. 1a). Each trial consisted of a sequence of two to eight stimuli after which participants provided a response

regarding the hidden state being tracked (Fig. 1b). In the cue-based (Cb) condition, participants were instructed to monitor the deck (category A or B) from which presented cards (oriented stimuli) are drawn (Fig. 1c). In the outcome-based (Ob) condition, the same participants were instructed to select the action (the left or right key press), which draws cards from a target deck (counterbalanced across blocks). As indicated above, the hidden state (the drawn deck in the Cb condition, or the target deck-drawing action in the Ob condition) reversed occasionally and unpredictably between trials, thereby requiring participants to adapt their behavior following each reversal.

Importantly, this experimental design allowed for an exact match of all task parameters (including stimulus characteristics, response parameters, and presentation times) and all computational variables (predicted by Bayesian inference) between Cb and Ob conditions. In particular, the amount of information provided by each stimulus sequence was strictly identical between conditions (Supplementary Fig. 1a; see "Methods" for analytical derivations). Their key difference is that the drawn category was independent of participants' previous response in the Cb condition, whereas it depended on participants' previous response (e.g., A if left or B if right) in the Ob condition.

**Slower reversal learning during outcome-based inference**. Neither response accuracy—i.e., the fraction of responses consistent with the hidden state (Cb: $81.7 \pm 0.7\%$, Ob: $81.9 \pm 0.8\%$, mean ± SEM, paired $t$-test, $t_{23} = 0.3$, $p = 0.784$)—nor mean response times (Cb: $594.3 \pm 51.6$ ms, Ob: $585.3 \pm 55.5$ ms, $t_{23} = -0.5$, $p = 0.648$) differed significantly across conditions. Despite this match, we found a selective difference in reversal learning between Cb and Ob conditions (Fig. 2a). Fitting response reversal curves by a saturating exponential function revealed a longer reversal time constant in the Ob condition (Fig. 2b; Cb: $0.82 \pm 0.10$, Ob: $1.28 \pm 0.12$, $t_{23} = 8.0$, $p < 0.001$), as well as a higher asymptotic reversal rate (Cb: $86.2 \pm 1.2\%$, Ob: $90.8 \pm 1.2\%$, $t_{23} = 5.2$, $p < 0.001$).

To characterize the origin of these learning differences, we constructed response repetition curves—i.e., the probability of repeating a previous response as a function of the evidence provided by the intervening sequence in favor of the previous response (see "Methods"). Positive evidence indicates evidence consistent with the previous response, whereas negative evidence indicates evidence conflicting with the previous response. Plotting these response repetition curves revealed a clear leftward shift in the Ob condition (Fig. 2c). Fitting these curves using a sigmoid function showed a selective increase in the point of subjective equivalence (PSE)—i.e., the amount of conflicting evidence required to switch more often than repeat the previous response (Fig. 2d; Cb: $1.03 \pm 0.10$, Ob: $1.43 \pm 0.09$, $t_{23} = 7.7$, $p < 0.001$). Neither the slope of response repetition curves—indexing the sensitivity of responses to evidence (Cb: $2.14 \pm 0.16$, Ob: $2.10 \pm 0.17$, $t_{23} = -0.2$, $p = 0.846$)—nor their lower asymptote—reflecting the small fraction of evidence-independent repetitions (Cb: $3.1 \pm 0.9\%$, Ob: $6.8 \pm 1.5\%$, $t_{23} = 2.0$, $p = 0.056$)—differed significantly between conditions. Together, these results indicate slower reversal learning in the Ob condition, caused by a larger amount of conflicting evidence required to reverse the previous response. This increased PSE in the Ob condition was present (and equally strong) on the first trial following each reversal of the hidden state, where evidence maximally conflicts with the previous response and did not depend on the number of stimuli in the sequence (Supplementary Fig. 2).

**Lower perceived hazard rate during outcome-based inference**. The tight match between Cb and Ob conditions allowed us to model human behavior in the two conditions by the same

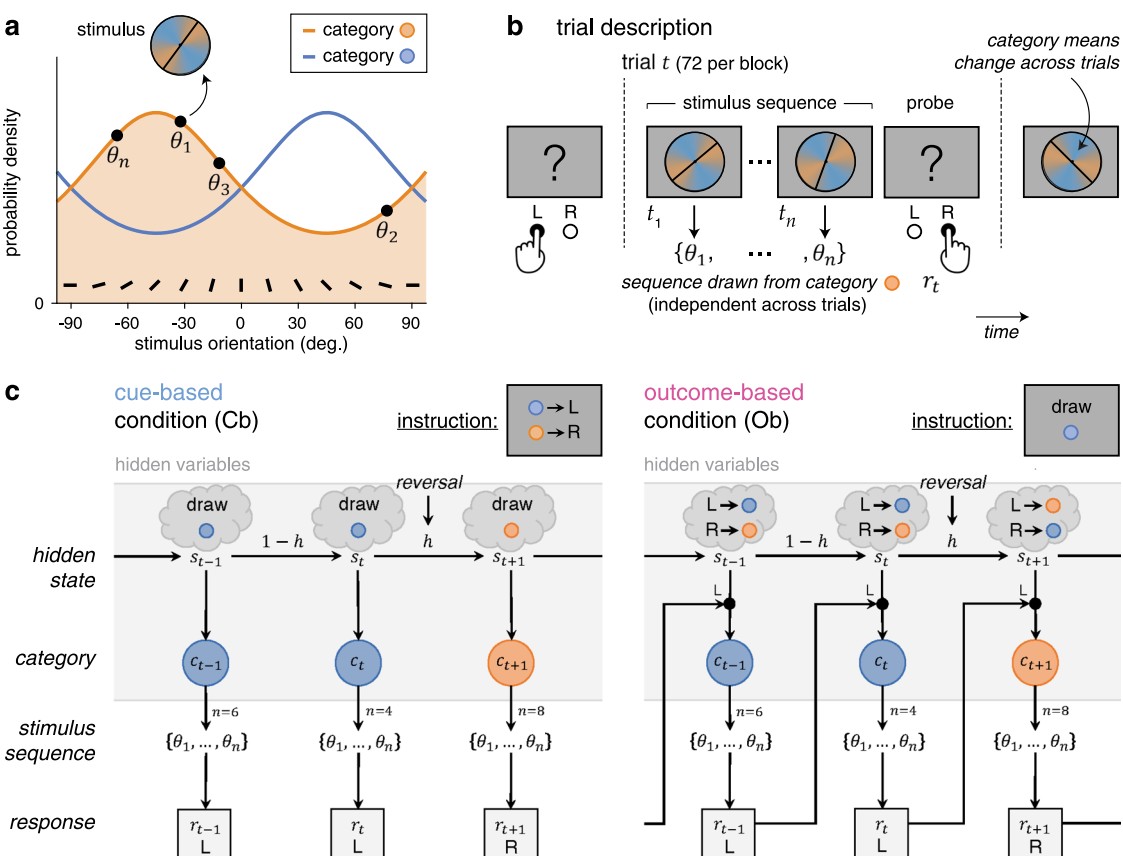

**Fig. 1 Reversal learning task and conditions. a** Generative stimulus distributions used for the two categories A and B. Stimuli correspond to bars of orientations $\{\theta_1, \dots, \theta_n\}$ drawn from one of two color-labeled categories, associated with overlapping probability density functions centered on orthogonal orientations. **b** Trial description. Each trial $t$ consists of a sequence of two to eight stimuli $\{\theta_1, \dots, \theta_n\}$ drawn from one of the two categories, presented at an average rate of 2 Hz, after which participants provide a response $r_t$ (the left or right key press) regarding the hidden state of the task being tracked. **c** Graphical description of cue-based and outcome-based conditions. Left: in the cue-based (Cb) condition, participants are instructed to monitor the deck (category A or B) from which presented cards (oriented stimuli) are drawn. This hidden state $s_t$ of the task alternates occasionally and unpredictably between trials. In this example, the hidden state $s_t$ reverses at the end of trial $t$. Right: in the outcome-based (Ob) condition, the same participants are instructed to select the action (the left or right key press), which draws cards from a target deck. The deck drawn at trial $t$ depends not only on the hidden state $s_t$ but also on the response $r_{t-1}$ provided at the previous trial. As in the cue-based condition, the hidden state $s_t$ (the target deck-drawing action in this case) alternates occasionally and unpredictably between trials. In this example, the hidden state $s_t$ reverses at the end of trial $t$.

Bayesian inference process (Fig. 3a), controlled by two parameters as follows: (1) the perceived hazard rate $h$—i.e., the subjective rate of reversals of the hidden state[15]—and (2) the inference noise $\sigma$—i.e., the SD of internal noise in the processing of each stimulus sequence[17]. On each trial, the model updates its belief regarding the current value of the hidden state by combining its prior belief, before seeing the new sequence of stimuli, with the evidence provided by each stimulus in the new sequence (see "Methods"). Given that the hidden state $s_t$ alternates between two possible values $s_1$ and $s_2$, beliefs can be expressed as log-odds, $\log\big(p(s_t = s_1)/p(s_t = s_2)\big)$: the sign of the log-odds belief indicates whether $s_1$ or $s_2$ is more likely, whereas the magnitude of the log-odds belief indicates the strength of the belief in favor of the more likely hidden state. Each update of the (log-odds) belief is corrupted by internal noise of SD $\sigma$. A fraction of the resulting posterior belief is then carried over as prior belief for the next trial as a function of the perceived hazard rate $h$: the larger the perceived hazard rate, the smaller the prior belief at the beginning of the next trial.

To quantify the (sub)optimality of human performance, we first simulated responses from the Bayesian inference model with optimal parameters[15]—i.e., the true hazard rate (0.125) and exact (noise-free) inference. The response accuracy of the optimal model (88.4 ± 0.3%), identical by construction across the two conditions, substantially exceeded human performance (Cb: $t_{23} = 9.3$, $p < 0.001$, Ob: $t_{23} = 7.8$, $p < 0.001$). To characterize the origin of human suboptimality and the nature of observed differences between Cb and Ob conditions, we then fitted the perceived hazard rate and the amount of inference noise to human behavior in each condition using particle Monte Carlo Markov Chain (MCMC; see "Methods").

Consistent with previous work[17], group-level analyses showed a substantial amount of inference noise, which did not differ across conditions (Fig. 3b; Cb: 0.512 ± 0.024, Ob: 0.550 ± 0.033, $t_{23} = 1.5$, $p = 0.141$). Beyond inference noise, we ruled out the presence of a stochastic "softmax" choice policy in both conditions by Bayesian model selection (BMS; Supplementary Fig. 1b–d; Cb: exceedance $p < 0.001$, Ob: exceedance $p < 0.001$). Together, these results suggest that human suboptimality arises in both conditions from inference noise, whose magnitude correlated significantly between conditions across participants (Pearson's $r = 0.630$, d.f. = 22, $p = 0.001$). Plotting participants' psychophysical kernels (Supplementary Fig. 1e) confirmed that most of the inference noise reflects genuinely random variability rather than stereotyped (e.g., leaky) inference (Supplementary Fig. 1f)—consistent again with previous work[17].

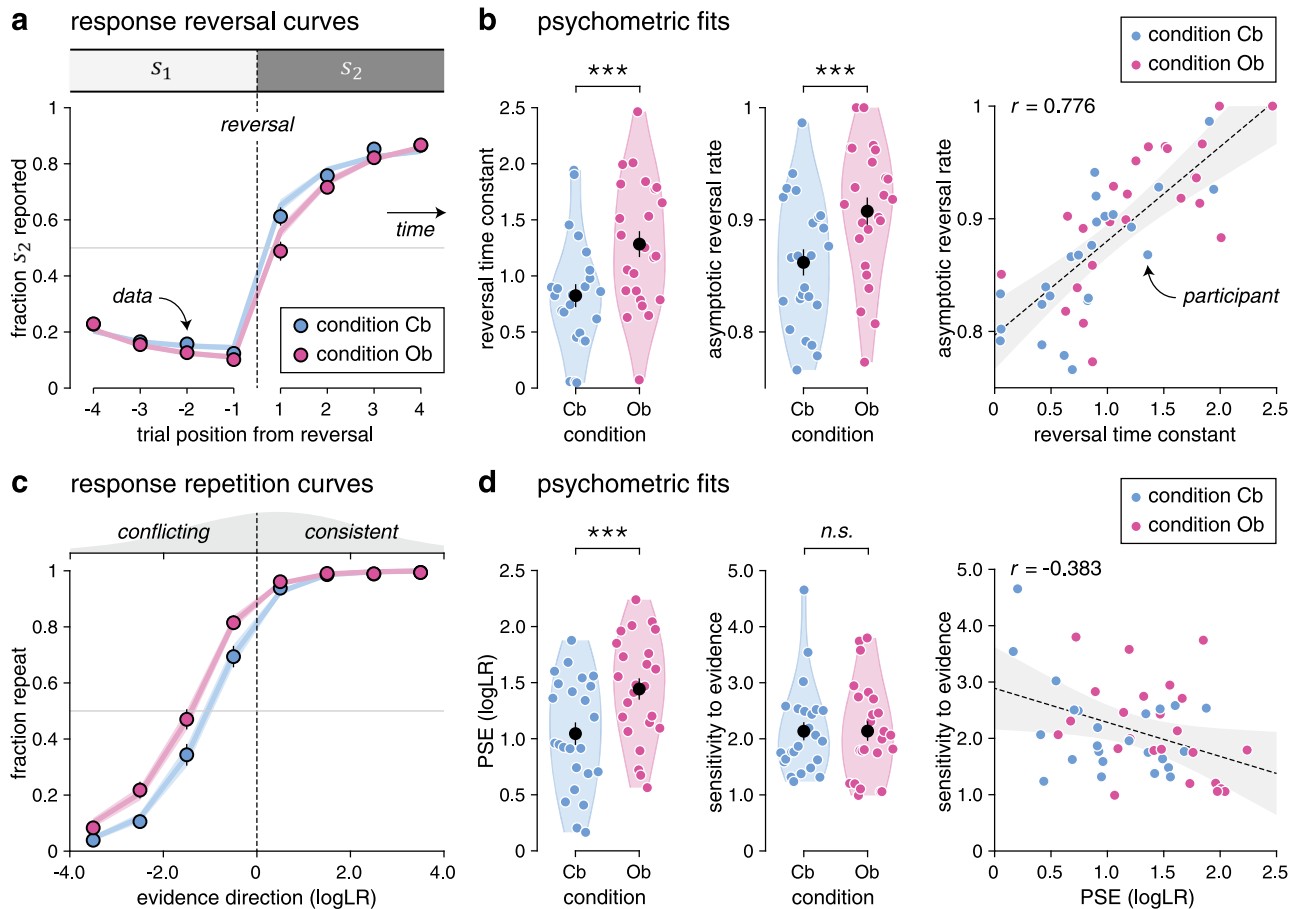

**Fig. 2 Reversal learning behavior and psychometric fits. a** Response reversal curves. Fraction of hidden state $s_2$ reported behaviorally (y axis) in the four trials preceding (left) and following (right) a reversal from $s_1$ to $s_2$ (x axis). The thin dotted line indicates the position of the reversal. Dots indicate the observed data (means ± SEM, n = 24 participants), whereas lines and shaded error bars indicate best-fitting saturating exponential functions (means ± SEM, n = 24 participants). **b** Best-fitting parameters of saturating exponential functions in the cue-based and outcome-based conditions. Black dots and error bars indicate group-level means ± SEM, whereas colored dots indicate participant-level estimates (n = 24 participants). Left: the reversal time constant is longer and the asymptotic reversal rate is higher in the outcome-based condition. Right: the two parameters correlate positively across participants. The thin dotted line indicates the best-fitting regression line and the shaded area its 95% confidence interval. **c** Response repetition curves. Fraction of response repetitions (y axis) as a function of the evidence provided by the intervening sequence in favor of the previous response (x axis, expressed as logLR). Positive evidence indicates evidence consistent with the previous response, whereas negative evidence indicates evidence conflicting with the previous response. The thin dotted line indicates perfectly uncertain (null) evidence. Dots indicate the observed data (means ± SEM, n = 24 participants), whereas lines and shaded error bars indicate best-fitting sigmoid functions (means ± SEM, n = 24 participants). **d** Best-fitting parameters of sigmoid functions in the cue-based and outcome-based conditions. Left: the PSE is increased in the outcome-based condition, whereas the sensitivity to evidence is matched across conditions. Right: the two parameters correlate slightly negatively across participants. Significant effect at ***$p < 0.001$; n.s., a nonsignificant effect (paired two-sided t-tests, d.f. = 23, no correction for multiple comparisons). The error band indicates the 95% confidence interval for the regression line fitted using ordinary least squares. Source data are provided as a Source Data file.

By contrast, comparing perceived hazard rates between conditions revealed a significant decrease in the Ob condition (Fig. 3c; Cb: 0.191 ± 0.022, Ob: 0.115 ± 0.015, $t_{23} = -7.7$, $p < 0.001$). This difference offers a computational account for the increased PSE in the Ob condition. Indeed, the decrease in perceived hazard rate in the Ob condition boosts prior beliefs by 42% (Fig. 3d), thereby requiring more conflicting evidence to reverse a previous response. Additional analyses confirmed that a larger fraction of the posterior belief is carried over to the next trial in the Ob condition, consistent with a lower perceived hazard rate but at odds with an increased response repetition bias (Supplementary Fig. 3).

This difference in perceived hazard rate between conditions makes a testable prediction: participants should be more accurate in the Ob condition in more stable environments where reversals are rare and more accurate in the Cb condition in more volatile

environments where reversals are frequent (Fig. 3e). Unbeknownst to participants, we varied the true hazard rate across blocks between 0.083 (more stable) and 0.167 (more volatile). As predicted, participants were more accurate in the Ob condition in more stable blocks ($t_{23} = 3.4$, $p = 0.002$) and more accurate in the Cb condition in more volatile blocks ($t_{23} = -2.5$, $p = 0.020$, interaction: $F_{1,23} = 20.2$, $p < 0.001$). This interaction is driven by the fact that participants did not adapt their perceived hazard rate to these fine, uncued changes in true hazard rate (Supplementary Fig. 4a, b), with perceived hazard rates closer to the true hazard rate for the Ob condition in more stable blocks and for the Cb condition in more volatile blocks. Importantly, the decrease in perceived hazard rate in the Ob condition was highly similar for participants with more stable inference (i.e., low perceived hazard rates) and participants with more volatile inference (i.e., high perceived hazard rates) across conditions (Supplementary Fig. 4c, d). This pattern of

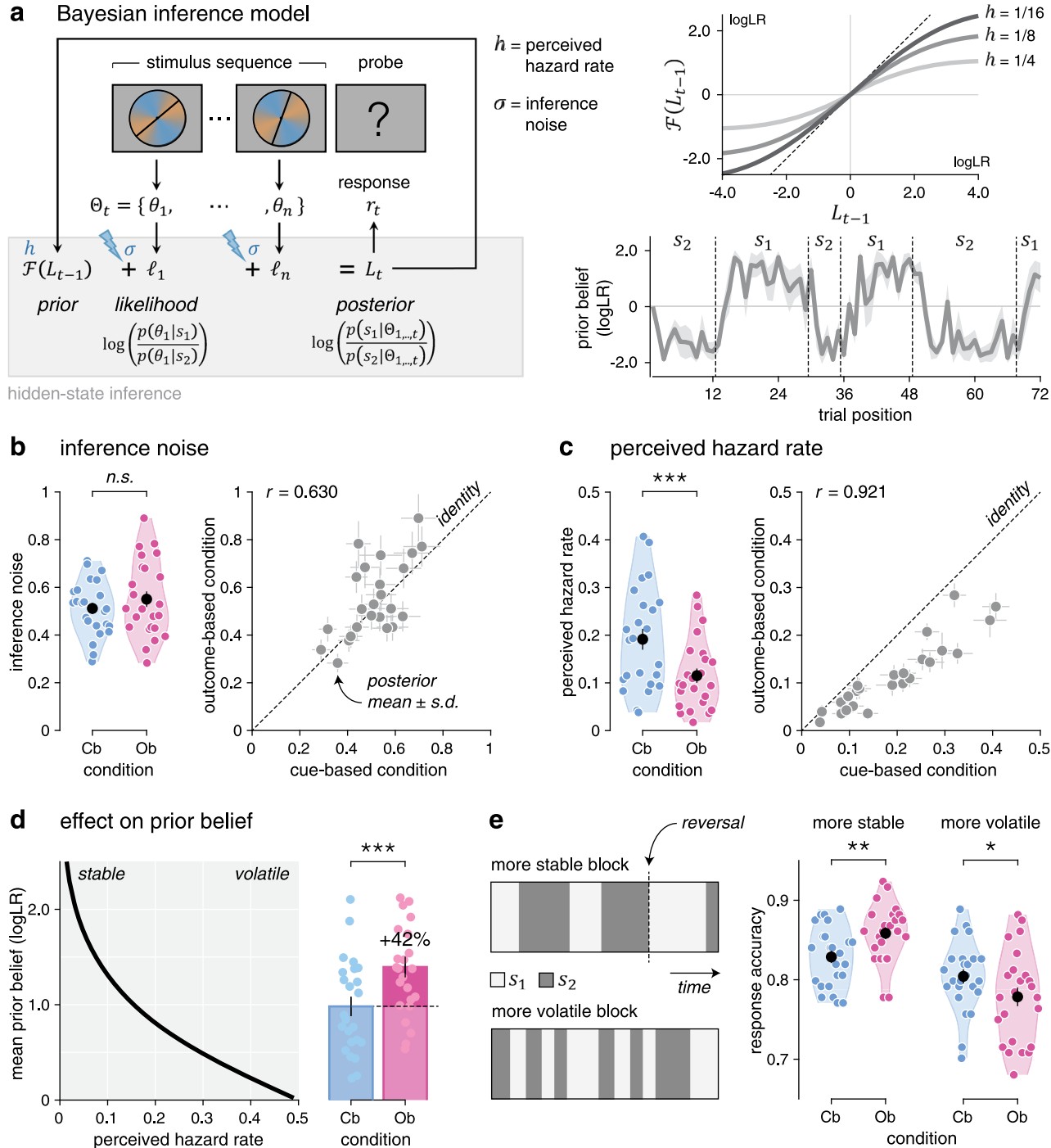

findings supports the conclusion that participants perceive the Ob condition as more stable than the Cb condition, despite identical true hazard rates.

**No difference between prospective and retrospective cue-based inference.** The Cb and Ob conditions differ in the degree of control conferred to participants over the sampling of information (instrumental control in the Ob condition, no control in the Cb condition), but not only. Indeed, outcome-based inference concerns the consequences of an action, a form of "prospective" inference, whereas cue-based inference concerns the cause of presented stimuli, a form of "retrospective" inference. To determine whether observed differences are due to the prospective

nature of inference in the Ob condition (rather than the control conferred to participants), we tested another group of participants ($N = 25$) in two variants of the Cb condition as follows: (1) a "retrospective" condition in which participants were asked to report the category from which the previous stimulus sequence was drawn and (2) a "prospective" condition in which the same participants were asked to report the category from which the next stimulus sequence will be drawn. Unlike the Ob condition, the prospective condition does not confer any instrumental control to participants. However, similar to the Ob condition, this new condition requires performing inference about the upcoming stimulus sequence.

The contrast between retrospective and prospective conditions did not yield significant differences in reversal learning

**Fig. 3 Computational modeling of behavior. a** Left: graphical description of the Bayesian inference model, controlled by the perceived hazard rate $h$ and the inference noise $\sigma$. On each trial, the model updates its belief $L$ regarding the current value of the hidden state $s_t$ by combining its prior belief $\mathcal{F}(L_{t-1})$ with the likelihood $\mathcal{L}_t$ associated with the stimuli in the new sequence, resulting in the posterior belief $L_t$. Right top: modeled fraction of the posterior belief at the end of trial $t-1$ ($x$ axis) carried over as prior belief at the beginning of trial $t$ ($y$ axis), for three values of perceived hazard rate. The larger the perceived hazard rate, the smaller the prior belief at the beginning of the next trial. Right bottom: simulated trajectory of the prior belief at the beginning of each trial over the course of an example block with $h = 1/8$ and $\sigma = 0.5$. The line and shaded error bar indicate the mean and SD across simulations. Vertical dotted lines indicate reversals of the hidden state. **b** Left: inference noise estimates in the cue-based and outcome-based conditions ($n = 24$ participants). Right: correlation between inference noise estimates in cue-based ($x$ axis) and outcome-based ($y$ axis) conditions ($n = 24$ participants). Dots and error bars indicate posterior means ± SD obtained by model fitting. The thin dotted line shows the identity line. Inference noise estimates do not differ between conditions. **c** Left: perceived hazard rate estimates in the cue-based and outcome-based conditions ($n = 24$ participants). Right: correlation between perceived hazard rate estimates in cue-based and outcome-based conditions. The perceived hazard rate is significantly lower in the outcome-based condition. Source data are provided as a Source Data file. **d** Left: predicted effect of perceived hazard rate ($x$ axis) on the magnitude of prior beliefs ($y$ axis). Dots and error bars indicate means ± SEM. Right: predicted magnitude of prior beliefs in the cue-based and outcome-based conditions. Bars and error bars indicate means ± SEM ($n = 24$ participants). **e** Effect of volatility on the behavioral difference between conditions. Left: evolution of the hidden state $s$ over the course of a more stable block (top) and a more volatile block (bottom). Right: response accuracy (data from $n = 24$ participants) is higher for the outcome-based condition in more stable blocks (left), but worse in more volatile blocks (right). Significant effect at $*p < 0.05$, $**p < 0.01$, $***p < 0.001$; n.s., a nonsignificant effect (paired two-sided $t$-tests, d.f. = 23, no correction for multiple comparisons).

(Supplementary Fig. 5a). Unlike the contrast between Cb and Ob conditions, participants showed nearly identical reversal time constants in the two conditions (Supplementary Fig. 5b; retrospective: $0.676 \pm 0.100$, prospective: $0.678 \pm 0.131$, $t_{24} = 0.1$, $p = 0.986$). Plotting the associated response repetition curves confirmed the absence of leftward shift in the prospective condition (Supplementary Fig. 5c). Importantly, the amount of conflicting evidence required to switch did not differ between these two conditions (Supplementary Fig. 5d; retrospective: $0.852 \pm 0.144$, prospective: $1.061 \pm 0.144$, $t_{24} = 1.7$, $p = 0.109$). The absence of difference between retrospective and prospective inference in the absence of instrumental control supports the notion that it is the presence of instrumental control conferred to participants in the Ob condition—not the prospective nature of outcome-based inference—which triggers the effects described above.

**Neural correlates of stimulus processing across space and time.** To identify the neural representations and dynamics supporting cue-based and outcome-based inference, we described each stimulus (1440 per condition and per participant) by a set of distinct characteristics: its orientation, its change (tilt) from the preceding stimulus, and the strength of the evidence it provides to the inference process. We then applied multivariate pattern analyses to MEG signals aligned to stimulus onset, to estimate the neural patterns, or "codes," associated with these characteristics. Owing to the fine temporal resolution of MEG signals and the absence of correlation between these characteristics, we could extract the time course of neural information processing within the first hundreds of milliseconds following stimulus onset (see "Methods").

The neural coding of stimulus orientation (Fig. 4a) peaked at 120 ms following stimulus onset (jackknifed mean, Cb: 123.1 ms, Ob: 122.4 ms), with equal precision across Cb and Ob conditions. The neural coding of stimulus change (Fig. 4b)—defined as the absolute tilt between the current stimulus and its predecessor in the sequence—peaked at 220 ms following stimulus onset (Cb: 225.7 ms, Ob: 223.3 ms), again with equal precision across conditions. The neural coding of stimulus evidence (Fig. 4c)—defined as the absolute tilt between the current stimulus and the nearest category boundary—peaked around 360 ms following stimulus onset (Cb: 357.5 ms, Ob: 378.7 ms). Unlike the two previous characteristics, the computation of stimulus evidence requires the mapping of stimulus orientation onto category-defined axes, which we varied randomly across trials. Despite its

relevance for inference, the coding precision of stimulus evidence did not differ between Cb and Ob conditions.

The neural codes of all three characteristics in MEG signals were highly dynamic: their associated cross-temporal generalization matrices showed sharp diagonals (Supplementary Fig. 6). Furthermore, the neural code of each characteristic progressed along mostly "null" dimensions for the other two characteristics, indicating low interference between the coding of the three characteristics (Supplementary Fig. 7).

To assess the degree of similarity of each neural code between Cb and Ob conditions, we performed cross-condition generalization analyses: we used the coding weights estimated in one condition to compute neural predictions in the other condition (see "Methods"). We also devised a procedure for estimating the degree of similarity between two neural codes based on a population of linear coding units (Fig. 5a). Applying this procedure to the neural codes of each stimulus characteristic indicated near-perfect similarity between Cb and Ob conditions—i.e., no loss in coding precision when using coding weights estimated in one condition to compute neural predictions in the other condition (Fig. 5b). These results suggest that stimulus processing in the Cb and Ob conditions relies on shared neural processes.

Next, we sought to identify the cortical distribution of these shared neural codes across conditions. For this purpose, we estimated the cortical sources of observed MEG signals and performed focal multivariate pattern analyses in source space using a "searchlight" approach (see "Methods"). Due to the linear mixing of sensor data used to reconstruct activity at each cortical source, we could decode each stimulus characteristic better than chance at all cortical sources (Supplementary Fig. 8). However, the coding precision of the same characteristic varied greatly across cortical sources and in different ways for the three characteristics. Therefore, to increase the spatial selectivity of obtained results, we estimated the cortical sources for which the coding precision of one stimulus characteristic significantly exceeds the coding precision of the other two characteristics (Supplementary Fig. 9). This analysis of selectivity identified a cluster in early visual cortex for stimulus orientation, peaking at 90 ms following stimulus onset (Fig. 4d). For stimulus change, the same analysis isolated clusters in the temporo-parietal junction and the posterior parietal cortex (PPC), peaking at 190 ms following stimulus onset (Fig. 4e). Finally, the coding of stimulus evidence was associated with clusters in the PPC and the lateral prefrontal cortex (LPFC) after 400 ms following stimulus onset (Fig. 4f). Together, these results highlight a spatiotemporal

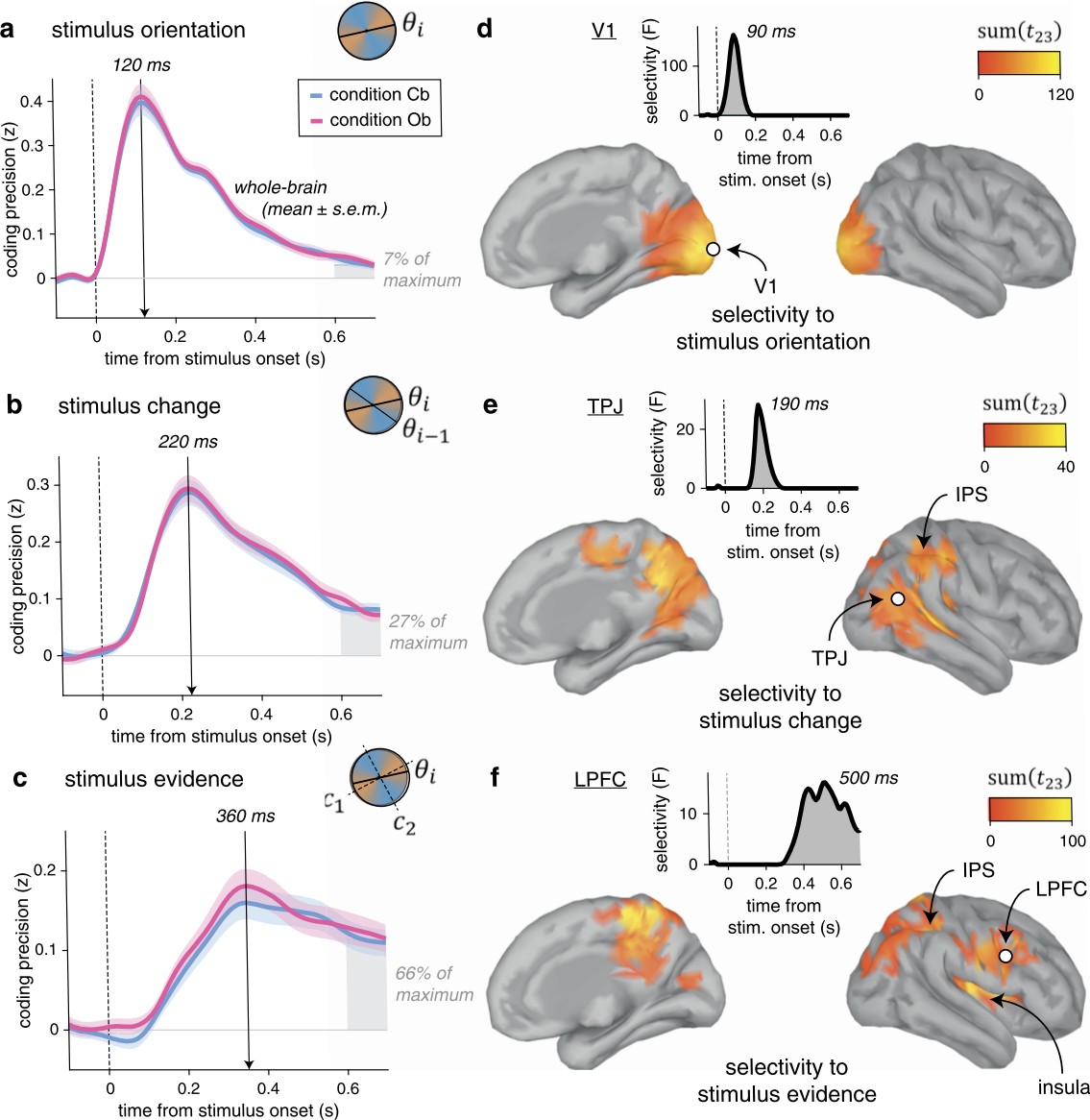

**Fig. 4 Neural correlates of stimulus processing. a** Time course of neural coding of stimulus orientation from whole-brain MEG signals. Stimulus orientation coding peaks at 120 ms following stimulus onset, with equal precision across conditions. Lines and shaded error bars indicate means ± SEM. **b** Time course of neural coding of stimulus change. Stimulus change coding peaks at 220 ms following stimulus onset, with equal precision across conditions. **c** Time course of neural coding of stimulus evidence. Stimulus evidence peaks at 360 ms following stimulus onset, with equal precision across conditions. The shaded area at 600–700 ms is used as reference for the three stimulus characteristics to compute their relative coding precision expressed as fraction of their respective maximum value. **d** Searchlight-based coding of stimulus orientation across the cortical surface (left: medial view, right: lateral view). The bilateral spatiotemporal cluster found in early visual cortex (including V1), marginalized across time (cluster definition $p = 0.001$, cluster statistic $p < 0.001$) corresponds to cortical sources for which the coding precision of stimulus orientation significantly exceeds the coding precision of stimulus change and stimulus evidence. Inset: time course of the selectivity statistic for a cortical source in V1 (white dot). **e** Searchlight-based coding of stimulus change. Change-selective clusters are found in the TPJ and the PPC. Inset: time course of the selectivity statistic for a cortical source in the TPJ (white dot). **f** Searchlight-based coding of stimulus evidence. Evidence-selective clusters are found in the PPC, the LPFC, and the insula. Inset: time course of the selectivity statistic for a cortical source in the LPFC (white dot). Source data are provided as a Source Data file.

cascade of neural patterns involved in the processing of each stimulus, shared across the two conditions.

**Neural correlates of the interaction between evidence and beliefs.** On each trial, inference requires updating the prior belief in the current value of the hidden state (the drawn category in the Cb condition or the target-drawing action in the Ob condition) with the evidence provided by each stimulus in the new sequence (Fig. 3a). By isolating the neural patterns involved in this

interaction between incoming evidence and prior beliefs, we sought to identify the brain signatures of the stabilization of hidden-state inference in the Ob condition.

First, we asked how well we could decode from MEG signals the evidence provided by each stimulus in favor of the ongoing belief at stimulus onset. For this purpose, we estimated in each condition the belief trajectory over the course of each sequence based on the following: (1) the evidence provided by each stimulus in the sequence and (2) best-fitting values for the perceived hazard rate $h$ and the amount of inference noise

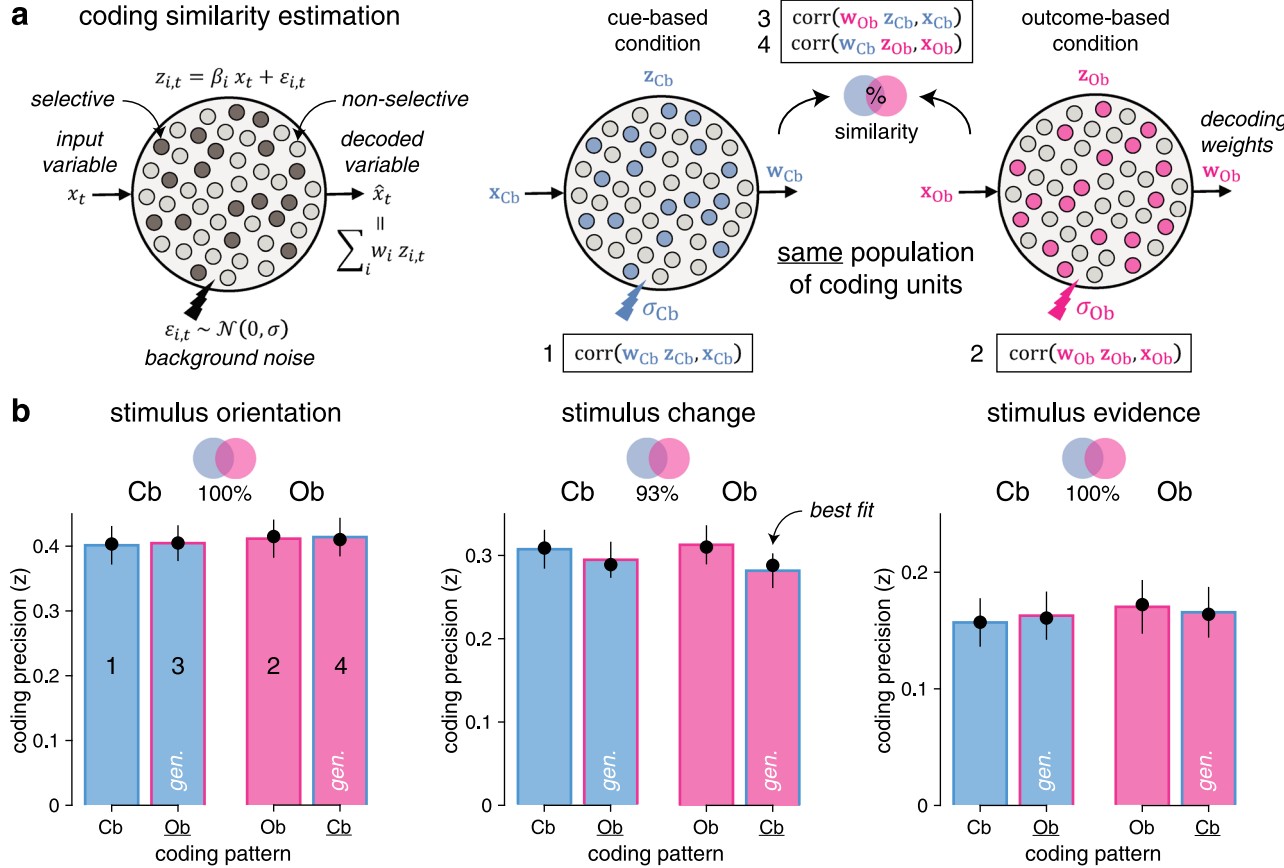

**Fig. 5 Estimation of neural coding similarity across conditions. a** Graphical description of the coding similarity estimation procedure. Left: a population of linear coding units $z_{i,t}$ represents an input scalar variable $x_t$ by a fraction of "selective' units ($z_{i,t} = \beta_i x_t + \varepsilon_{i,t}$ where $\beta_i \neq 0$) with background noise $\varepsilon_{i,t}$ of SD $\sigma$. A linear decoder is applied to compute an estimate $\hat{x}_t$ of the input variable from population activity. Right: the same population of coding units represents the same input variable $x$ by partially overlapping selective units $z$ in the cue-based (left) and outcome-based (right) conditions. Computing coding precision within each condition (1–2) and across conditions (3–4, marked by gen., by using the coding weights $w$ estimated in one condition to compute neural predictions in the other condition) allows to quantify the degree of similarity (overlap) between selective units in the two conditions. **b** Estimated coding similarities for stimulus orientation (left), stimulus change (middle), and stimulus evidence (right). Bars and error bars indicate jackknifed means ± SEM ($n = 24$ participants). Dots show predicted values obtained by simulating the population of coding units with best-fitting estimates of similarity and background noise. Bar fillings indicate the condition in which neural predictions are computed. Bar outlines indicate the condition in which coding weights are estimated. This procedure indicates near-perfect coding similarity between cue-based and outcome-based conditions for stimulus orientation (left, jackknifed mean: 100%), stimulus change (center, 93%), and stimulus evidence (right, 100%). Source data are provided as a Source Data file.

$\sigma$ (see "Methods"). Estimated belief trajectories thus accounted for the lower perceived hazard rate and the more stable beliefs found in the Ob condition (Fig. 6a). Positive evidence indicates evidence consistent with the ongoing belief, whereas negative evidence indicates evidence conflicting with the ongoing belief.

We could decode this "consistency" variable in both conditions over a large time window starting around 110 ms following stimulus onset (Fig. 6b), with no significant difference between conditions. Similar to other stimulus characteristics, the cross-condition generalization analysis of this neural code revealed near-perfect similarity across conditions (Supplementary Fig. 10a). Importantly, this consistency variable assumes that participants update their belief regarding the current value of the hidden state after each stimulus, throughout each sequence of stimuli—and not only when probed for a response (Supplementary Fig. 10b). Distinguishing between stimulus-level and response-level inference schemes is not possible from behavior alone, as their applications result in the same behavior. However, the two inference schemes are associated with different belief trajectories over the course of each sequence. In particular, stimulus-level inference predicts that changes of the mind can occur midway through a sequence, as soon as a conflicting

stimulus flips the sign of the log-odds belief. As a result, the two inference schemes are associated with different consistency variables, whose coding precision in MEG signals can be compared. This comparison revealed a stronger coding of consistency predicted by stimulus-level inference from 150 ms following stimulus onset (Supplementary Fig. 10c; peak $F_{1,23} = 28.2$, cluster-level $p < 0.001$). This result indicates that participants update their beliefs regarding the current value of the hidden state after each stimulus—at a sub-second timescale—not only when probed for a response.

**Delayed reversal of beliefs during outcome-based inference.** We reasoned that the neural coding of the consistency of evidence with ongoing beliefs offers leverage to validate the lower perceived hazard rate in the Ob condition. Instead of computing belief trajectories based on the hazard rate $h^*$ fitted to participants' behavior (which was thus lower in the Ob condition), we estimated separately in each condition and for each participant the "neural" hazard rate for which we could best decode consistency from MEG signals—ignoring its best-fitting value obtained from behavior (Fig. 6c). This analysis revealed a lower neural hazard rate in the Ob condition (Cb: $0.319 \pm 0.023$, Ob: $0.216 \pm 0.026$,

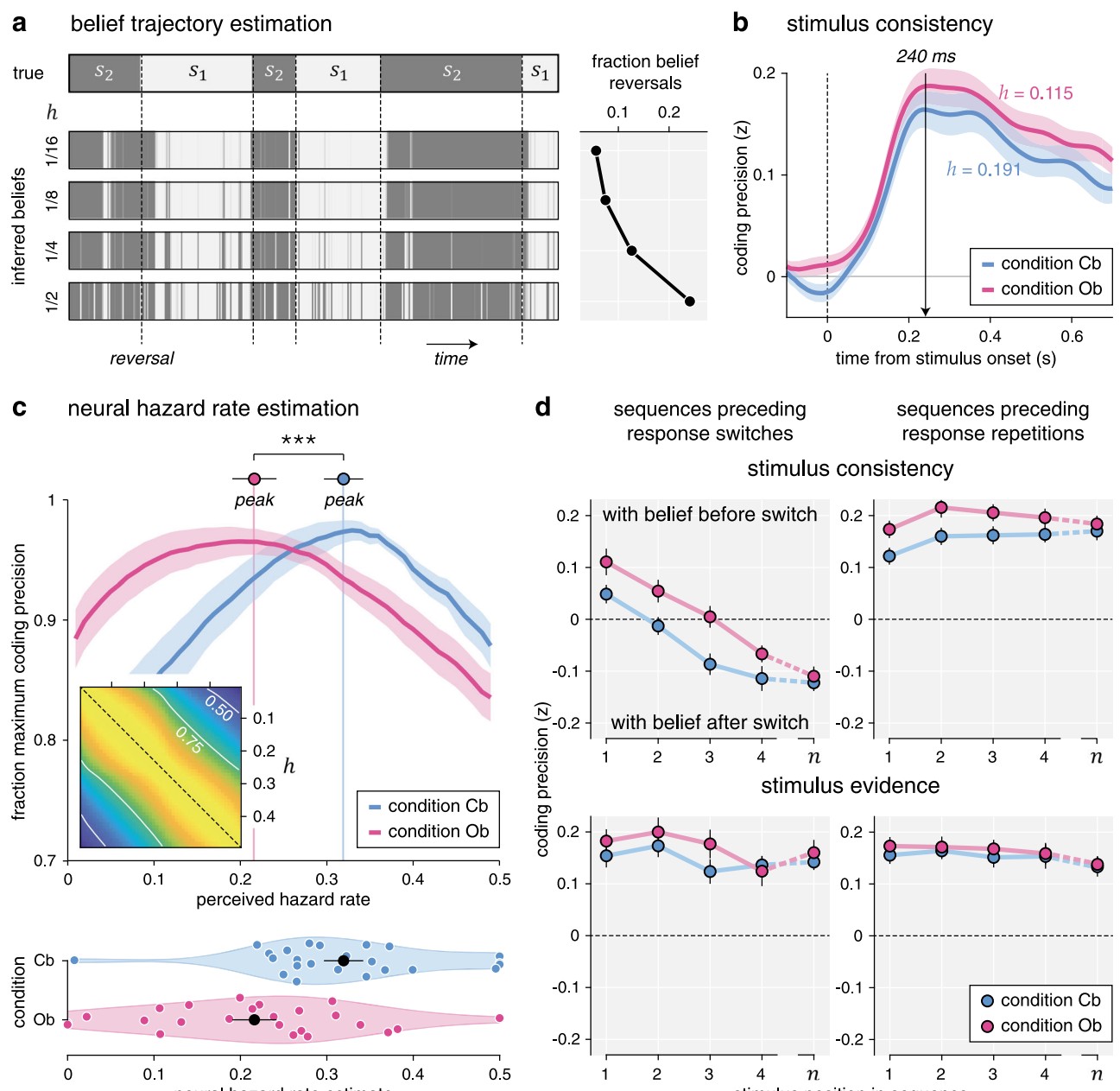

**Fig. 6 Belief trajectory estimation and analysis. a** Estimation of belief trajectory from behavior and perceived hazard rate. Left: each horizontal strip indicates the evolution of the hidden state *s* over the course of an example block. The upper strip indicates the true hidden state (with vertical dotted lines indicating reversals), whereas the lower strips indicate inferred beliefs in the current hidden state for increasing perceived hazard rates, from 1/16 (top) to 1/2 (bottom). Belief trajectories at trial *t* are conditioned on observed responses up to trial *t* using particle filtering. Right: fraction of mid-sequence belief reversals associated with each belief trajectory, growing with the perceived hazard rate. **b** Neural coding of consistency with the ongoing belief, computed using the estimated belief trajectory based on the best-fitting perceived hazard rate in each condition (lower in the outcome-based condition). Lines and shaded error bars indicate means ± SEM. Consistency is coded equally precisely in the two conditions. **c** Estimation of neural hazard rate. Relative coding precision of consistency (*y* axis, expressed as the fraction of the maximum per participant and condition) as a function of perceived hazard rate (*x* axis). The neural hazard rate is defined as the perceived hazard rate, which maximizes coding precision (dots and error bars indicate its mean ± SEM). Lines and shaded error bars indicate means ± SEM. Inset: correlation matrix ($r^2$) for consistency based on the full range of possible hazard rates. Estimation of the neural hazard rate relies on differences between consistency variables. **d** Neural coding trajectories preceding response reversals (left column) and repetitions (right column), for consistency with the belief at the beginning of the current sequence (top row) and the strength of stimulus evidence (bottom row). Sequences leading to response reversals are associated with switches in the neural coding of consistency, not stimulus evidence, which are delayed in the outcome-based condition. Dots and error bars indicate means ± SEM (*n* = 24 participants). Three stars indicate a significant effect at *p* < 0.001 (paired two-sided *t*-test, d.f. = 23, no correction for multiple comparisons). Source data are provided as a Source Data file.

$t_{23} = -4.5$, $p < 0.001$), thereby mirroring the lower perceived hazard rate found in this condition.

Finally, we conducted an additional analysis to estimate belief trajectories throughout a sequence from the neural coding of consistency, with the idea of visualizing more stable beliefs in the Ob condition. We reasoned that the neural coding of consistency, defined in relation to the prior belief at the beginning of the current sequence, should switch sign when the belief itself switches midway through the sequence. These mid-sequence belief switches are bound to occur in trials leading to response reversals (where the belief at the end of the trial has opposite sign to the belief at the beginning of the trial), but not in sequences leading to response repetitions (where the belief at the end of the trial has the same sign as the belief at the beginning of the trial). Furthermore, these coding switches should occur later inside a sequence in the Ob condition where prior beliefs are larger and thus require more conflicting evidence to switch sign. To test these different neural predictions, we decoded consistency at each stimulus position in a sequence (Fig. 6d), separately for sequences leading to response reversals and sequences leading to response repetitions. As response reversals are less frequent than response repetitions, we used in each condition a "common-filters" approach by applying the same coding weights for the two types of sequences (see "Methods").

Our predictions were corroborated by the neural data. First, sequences leading to response reversals were associated with coding switches: the first stimulus in the sequence was coded positively—i.e., in direction of the belief at the beginning of the trial (Cb: $0.049 \pm 0.018$, $t_{23} = 2.7$, $p = 0.011$, Ob: $0.111 \pm 0.026$, $t_{23} = 4.3$, $p < 0.001$)—whereas the last stimulus in the same sequence was coded negatively—i.e., in direction of the belief at the end of the trial (Cb: $-0.124 \pm 0.016$, $t_{23} = -7.7$, $p < 0.001$, Ob: $-0.111 \pm 0.019$, $t_{23} = -5.9$, $p < 0.001$). These coding switches were not observed during sequences leading to response repetitions: the last stimulus was also coded positively (Cb: $0.172 \pm 0.018$, $t_{23} = 9.6$, $p < 0.001$, Ob: $0.185 \pm 0.016$, $t_{23} = 11.5$, $p < 0.001$). Second, coding switches were significantly delayed in the Ob condition (jackknifed regression of zero-crossing times, $t_{23} = 4.6$, $p < 0.001$). Together, these findings support the presence of more stable beliefs in the Ob condition.

**Neural dissociation between absolute and relational coding of evidence.** Neural responses to each stimulus revealed two concurrent coding schemes of the evidence provided to the inference process as follows (Fig. 7a): (1) an "absolute" coding scheme reflected in the strength of the evidence and (2) a "relational" coding scheme reflected in the consistency of the same evidence with current beliefs about the hidden state. Examining their time courses (Fig. 7b) revealed a marked difference in coding dynamics: the relational (belief-dependent) coding scheme showed a faster rise in precision than the absolute (belief-independent) coding scheme, from 70 to 300 ms following stimulus onset (peak $t_{23} = 7.0$, cluster-level $p < 0.001$), and peaked about 120 ms earlier (jackknifed mean, relational: 243.4 ms, absolute: 367.6 ms, $t_{23} = -5.6$, $p < 0.001$).

A second fundamental difference between the two coding schemes concerns how their precision varies with the magnitude of beliefs (Fig. 7c). Indeed, the precision of the relational scheme scaled positively with the magnitude of beliefs (linear regression, Cb: $\beta = 0.052 \pm 0.008$, $t_{23} = 6.3$, $p < 0.001$, Ob: $\beta = 0.063 \pm 0.010$, $t_{23} = 6.7$, $p < 0.001$). It is expected from a coding scheme, which reflects the consistency of incoming evidence in relation to the current belief, and is thus not defined in the absence of beliefs. By contrast, the precision of the absolute scheme did not increase but rather decreased slightly with larger beliefs (Cb: $\beta = -0.013 \pm 0.005$, $t_{23} = -2.3$, $p = 0.031$,

Ob: $-0.017 \pm 0.004$, $t_{23} = -4.3$, $p < 0.001$), a pattern shared with the coding of stimulus orientation in early visual cortex (Supplementary Fig. 11a). Importantly, the precision of the absolute scheme did not depend on whether incoming evidence was consistent or conflicting with the current belief, in either condition (Supplementary Fig. 11b). In other words, the absolute coding scheme reflects the objective (veridical) amount of evidence provided by each stimulus, whereas the relational coding scheme reflects the interaction between the same evidence and subjective (inferred) beliefs.

Last, we identified cortical sources for which the precision of the relational coding scheme is significantly larger than the precision of the absolute coding scheme (Fig. 7d). Averaging across conditions revealed a bilateral temporal cluster (cluster-level $p < 0.001$), whose timing of effect matched the early rise of the relational coding of evidence at the whole-brain level (Supplementary Fig. 11c). These results support an active involvement of associative temporal regions, including the middle temporal gyrus and the entorhinal cortex, in the coding of incoming evidence in relation to ongoing beliefs—a coding scheme stabilized during outcome-based inference.

## Discussion

Accurate decision-making in uncertain environments requires identifying the generative cause of observed stimuli (cue-based inference, as in perceptual decisions), but also the expected consequences of one's own actions (outcome-based inference, as in reward-guided decisions). These two types of inference differ in the degree of control the decision-maker has over the sampling of evidence, a variable that is usually confounded with other task variables in paradigms used to study perceptual and reward-guided decisions. By comparing cue-based and outcome-based inference in tightly matched conditions, we show that interacting with uncertain evidence—rather than observing the same evidence—increases the perceived stability of volatile environments.

A first immediate conclusion is that the status of the decision-maker—an agent interacting with its environment in the Ob condition or an observer contemplating the same environment in the Cb condition—shapes human learning and decision-making under uncertainty. In the learning sciences, the distinction between "active" and "passive" information gathering has been stressed for decades[11]. Controlling the flow of incoming information during learning is seen as an important degree of freedom for acquiring non-redundant information and accelerating the learning process. By contrast, and surprisingly, existing theories of cognitive inference either do not consider the degree of control over the sampling of evidence as a contextual variable[2,4,7,15] or formulate distinct computational models of "active" and "passive" inference[18]. Our reversal learning task allows us to study the effect of control on inference by comparing two conditions that differ only in the instrumental control over the sampling of evidence. This selective difference changes the nature of uncertainty, but not the complexity of inference across conditions: uncertainty concerns the generative cause of stimuli in the Cb condition, whereas it concerns the consequence of the previous action in the Ob condition. We constructed the task such that the amount of uncertainty is perfectly matched across conditions: Bayes-optimal inference[15] results in the exact same belief trajectories and thus the exact same decision behavior. In other words, observed differences between conditions cannot be attributed to any difference in the evidence provided to participants. Despite this match, we found that interacting with uncertain evidence results in the following: (1) slower reversal learning, which decreases performance in volatile environments where reversals are frequent and

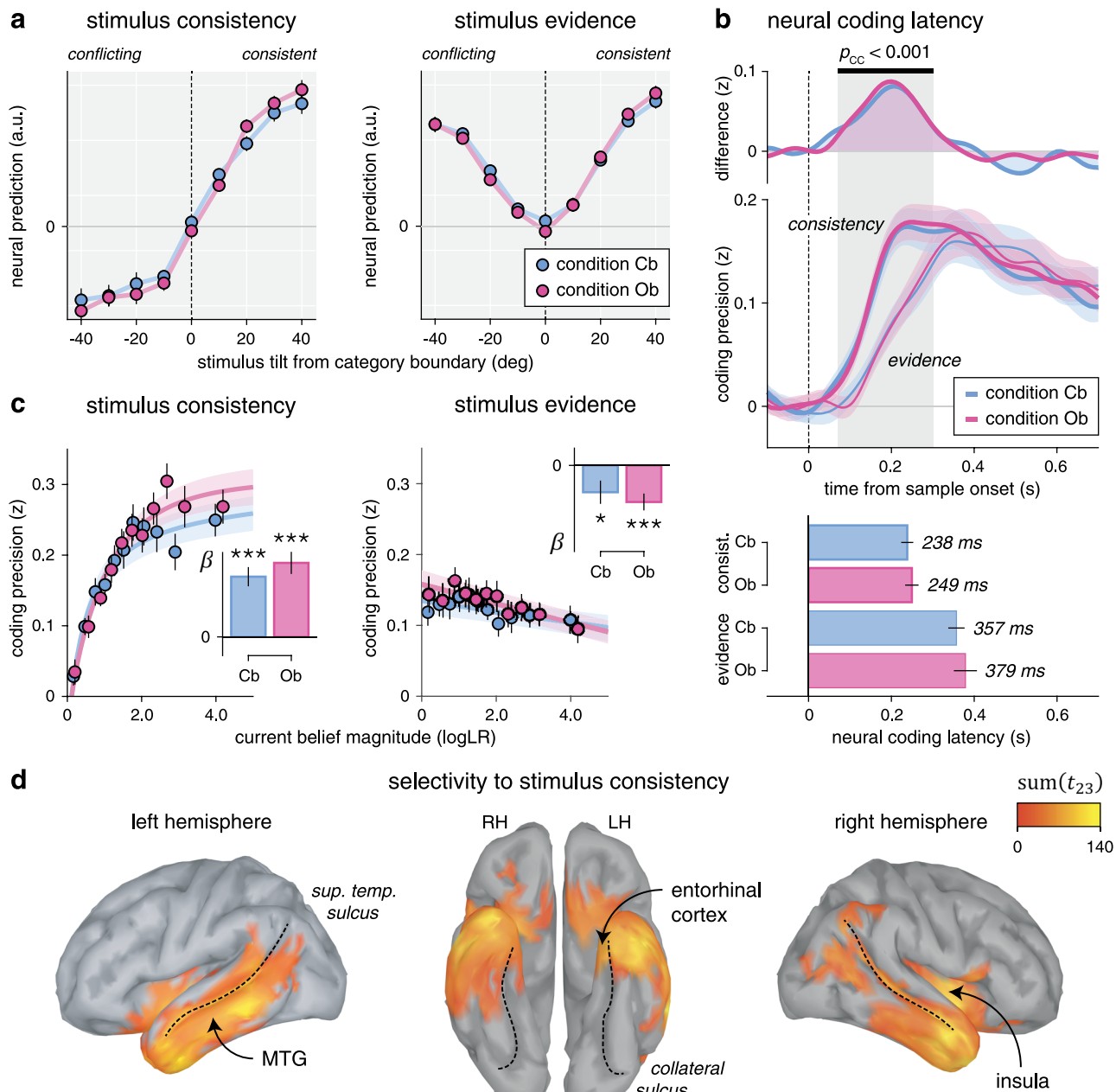

**Fig. 7 Neural correlates of absolute and relational coding of evidence. a** Neural predictions for stimulus consistency (left) and evidence (right), binned as a function of stimulus tilt from the nearest category boundary. Stimulus tilt is signed in relation to the current belief: a positive tilt indicates a stimulus consistent with the current belief, whereas a negative tilt indicates a stimulus conflicting with the current belief. Neural predictions of consistency (left) have opposite signs for belief-consistent and belief-conflicting evidence, whereas neural predictions of evidence strength (right) are independent of the current belief. Dots and error bars indicate means ± SEM ($n = 24$ participants). **b** Neural coding latencies for stimulus consistency and stimulus evidence. Top: neural coding time courses (lower panel) and their difference (upper panel). Stimulus consistency (thick lines) shows a faster rise in coding precision than stimulus evidence (thin lines). Lines and shaded error bars indicate means ± SEM. The shaded area indicates a significant cluster corrected for multiple comparisons at $p < 0.001$ based on paired two-sided $t$-tests. Bottom: estimated neural coding latencies. The neural coding of consistency peaks 120 ms earlier than the neural coding of evidence. Bars and error bars indicate jackknifed means ± SEM ($n = 24$ participants). **c** Effect of current belief magnitude on the neural coding of consistency (left) and evidence (right), grouped in equally sized bins. The coding precision of consistency (left), but not evidence (right), scales positively with belief magnitude. Dots and error bars indicate means ± SEM. Lines and shaded error bars show parametric fits. Inset: linear regression slope estimates (jackknifed means ± SEM) of coding precision against belief magnitude ($t$-test against zero, d.f. = 23, uncorrected for multiple comparisons). **d** Searchlight-based differences in coding precision between consistency and evidence across the cortical surface (cluster definition $p = 0.001$, cluster statistic $p < 0.001$). Thin dotted lines follow the superior temporal sulcus (left and right) and the collateral sulcus (middle). Source data are provided as a Source Data file.

(2) higher asymptotic reversal rates, which increases performance in stable environments where reversals are rare.

Interacting with uncertain environments confers several cognitive advantages, which could explain the lower perceived hazard rate and the resulting stabilization of hidden-state inference in the Ob condition. First, the active sampling of evidence enables testing current hypotheses in a self-directed, online manner—an ability known to improve human learning in novel environments[11,12]. A functional interpretation is that the lower perceived hazard rate in the Ob condition aims at testing the current hypothesis that the previous action samples from the target category. Such hypothesis testing is by definition impossible in the Cb condition, because participants had no control over the sampling of evidence. It is noteworthy that the inference process used in the Cb condition is not equivalent to "observational" learning—i.e., learning from or about another agent interacting with uncertain evidence[19,20]. Indeed, participants were actively committing to interpretations regarding the source of evidence in both conditions, the sole difference being the presence or absence of instrumental control over the sampling of evidence. Even without considering a hypothesis testing, knowledge about action-outcome contingencies affords an agent to predict incoming sensory signals and to stabilize these signals by stabilizing its behavior[18,21,22]. The lower perceived hazard rate in the Ob condition may thus aim at stabilizing the source of the incoming evidence—something which is not possible in the absence of control. Last, the difference in perceived hazard rate between the two conditions may betray a temporal hierarchy between two types of changes, which can arise in uncertain environments: (1) external (uncontrollable) changes in the state of the environment (as in the Cb condition) and (2) changes in the instrumental control of the same environment (as in the Ob condition). In this view, participants behave as if they assume changes in control to be less frequent than external changes occurring in equally uncertain environments[23–25].

Although we were primarily interested in differences between conditions, we also found striking similarities in the computations and neural correlates of cue-based and outcome-based inference. At the behavioral level, participants showed correlated inference parameters between conditions. In particular, inference was subject to the same limited precision[16,17,26] in the two conditions. These correlations indicate that the two types of inference rely to a large extent on shared cognitive processes. At the neural level, multiple stimulus properties could be decoded from MEG responses with the same precision in the two conditions, from stimulus orientation in early visual cortex to stimulus evidence in the PPC and LPFC[8,27]. By applying weights estimated in one condition to decode stimulus properties in the other condition, we further demonstrate that the underlying neural "codes" are shared across the two conditions.

The computational modeling of inference[15] accounted for behavioral differences by an increased reliance on prior beliefs in the Ob condition, as if participants perceived a lower rate of reversals than in the Cb condition. Several alternative accounts can be ruled out based on different aspects of the behavioral and neural data. First, participants did not rely more on prior beliefs in the Ob condition, because they perceived the incoming evidence as less reliable: they showed equal sensitivity to the presented evidence and were instructed explicitly that the same probability distributions (categories) were sampled in the two conditions. Furthermore, participants showed no sign of leaky inference within each sequence, in either condition. The slower reversal learning in the Ob condition is also unlikely to arise from a biased, choice-supportive filtering of evidence described and reported across cognitive domains[28–32]. Indeed, such "confirmation bias" (which filters out belief-inconsistent evidence)

predicts a decreased sensitivity to evidence in the Ob condition— something which we did not observe. Another observation that stands at odds with a confirmation bias is that the strength of belief-inconsistent evidence could be decoded from MEG signals with equal precision across conditions. Finally, the reliance on prior beliefs cannot be explained by a heuristic choice strategy, which samples actions either from the prior belief or from the incoming evidence without ever combining them[33,34]. Contrary to these accounts, we could decode from MEG signals the explicit interaction between these two terms: the consistency of incoming evidence with inferred beliefs.

This consistency signal reflects a "relational" coding of evidence in relation to the current belief about the hidden state being monitored. It is best decoded from MEG signals localized in the temporal lobe, where both spatial and non-spatial cognitive maps have been identified in recent years[35–38]. Examining the fine temporal dynamics of this neural code revealed that participants update their beliefs following each stimulus and thus not only when probed for a response. In other words, this coding scheme maps the current stimulus not in relation to the previous decision but in relation to the current belief. We show that this belief-dependent coding of evidence predicts changes of mind in the form of coding switches during the preceding sequence: the first stimuli are coded in relation to the belief at the beginning of the trial (before the change of mind), whereas the later stimuli are coded in relation to the belief at the end of the trial (after the change of mind). Importantly, the precision of this neural code scales with the magnitude and direction (sign) of beliefs, thus offering a neural window into their trajectories. The dynamics of this neural code confirmed the lower perceived hazard rate in the Ob condition. In particular, coding switches occurred later in the Ob condition during sequences leading to changes of the mind, as predicted by larger prior beliefs that require more conflicting evidence to switch.

In terms of computations, this relational code corresponds to a contextual, history-dependent variable, not to a momentary evidence variable independent of ongoing beliefs[7,39–41]. Indeed, its reference point (i.e., the current belief) can switch sign within a single sequence of stimuli. In the Ob condition, this means that a non-target stimulus is coded negatively at the beginning of a sequence, but another can be coded positively later in the same sequence if a change of mind has occurred between the two stimuli—unlike classical descriptions of reward prediction errors in the midbrain[42,43]. Such dynamic coding scheme is however consistent with recent accounts of opposite coding of factual and counterfactual outcomes in the frontal cortex[44], and of hidden-state inference by dopaminergic signals[45]. Another property of this relational code, shared with other prediction error signals, is that it scales with the magnitude of the current belief: it is almost not expressed when the current belief is weak, but strongly expressed when the current belief is strong. This consistency signal can thus be described as a dynamic prediction error, tied to the current belief and not to the previous decision, which can be used by canonical "temporal difference" learning algorithms[5,6] in both conditions to update the current belief following each new piece of evidence.

The increased stability of uncertain beliefs in the presence of control bears important consequences for understanding psychiatric dysfunctions of learning and decision-making under uncertainty. In particular, the description of obsessive-compulsive disorder (OCD) as a "doubting disease" fails to distinguish between uncertainty about the cause of external events and uncertainty about the consequence of one's own actions. OCD patients may show a selective impairment of outcome-based inference (when inference concerns the uncertain outcomes of their actions), but no alteration of cue-based inference

(when inference concerns the uncertain source of observed stimuli). Even in the general population, the perceived degree of control over uncertain environments may explain variations in the attitude of individual decision-makers to risk over short timescales (by underestimating the true volatility of uncertain environments in the presence of control), but also variations in the accuracy of collective decisions (by increasing the perceived conflict between beliefs across individuals when they concern the expected outcomes of their actions).

## Methods

**Participants**. Twenty-four adult participants took part in the main magnetoencephalography study contrasting cue-based and outcome-based inference (12 females, mean age: 24 years, age range: 20–30 years, all right-handed). Thirty additional adult participants took part in the control behavioral study contrasting retrospective and prospective Cb inference (5 excluded due to chance-level performance in more than 1 block of trials, 15 females in the retained sample, mean age: 26 years). Participants had no history of neurological and psychiatric disease, and had normal or corrected-to-normal vision. All tested participants gave a written informed consent before taking part in the study, which received ethical approval from relevant authorities (Comité de Protection des Personnes Ile-de-France VI, ID RCB: 2007-A01125-48, 2017-A01778-45). Participants received a monetary compensation of 80 euros for their participation in the main magnetoencephalography study or 15 euros plus an additional bonus between 5 and 10 euros depending on their performance for their participation in the control behavioral study.

**Experimental task**. We asked participants to play a reversal learning task based on visual stimuli, which we framed in two conditions corresponding to different blocks of trials, either as cue-based or outcome-based inference. In both conditions, participants were asked to track a hidden state $s$ of the task, which alternates occasionally and unpredictably between two discrete values. Stimuli corresponded to oriented black bars presented in the foreground of a colored disc displaying an angular gradient between orange and blue (through gray)—the two cardinal colors being spaced by $\pi/2$. On each trial, a sequence of 2, 4, 6, or 8 stimuli was drawn from a von Mises probability distribution (category $c$) centered either on the orientation indicated by orange (A) or the orientation indicated by blue (B) with a fixed concentration $\kappa$ of 0.5 (Fig. 1a). The number of stimuli in each sequence was sampled pseudo-randomly and uniformly across trials. Each sequence was presented at an average rate of 2 Hz, using an inter-stimulus interval of $500 \pm 50$ ms. The last stimulus of each sequence was followed by a longer delay of $1000 \pm 50$ ms, before a change in the fixation point probed the participant for a response, by pressing either of two keys with their left or right index finger. All stimuli were presented on a gray background, at a viewing distance of 80 cm. Visual fixation was monitored online throughout the main task using an EyeLink 1000 eye-tracking system (SR Research, Ottawa, Canada), using a monocular tracking of the dominant eye at a sampling frequency of 1000 Hz.

Each trial consisted of a stimulus sequence followed by a response (Fig. 1b). The task was divided in 8 blocks of 72 trials (~7 min each). At the beginning of each block, an instruction screen framed the upcoming block either as an "observation" (Cb) block or as an "action" (Ob) block. The task was presented to participants as a card game, each stimulus depicting a card whose color is determined by the orientation of the black bar relative to the colored background. Each series of cards could be drawn either from the orange deck or from the blue deck, corresponding to the categories A and B. Participants were instructed explicitly that the decks were partially shuffled, such that the orange deck contained mostly orange cards but also blue cards and the blue deck contained mostly blue cards but also orange cards. We also exposed participants to $n = 20$ draws from each deck (each draw being sampled from a von Mises distribution using the fixed concentration $\kappa$ of 0.5 used in the main task) before practicing the task.

The description of the "observation" (Cb) and "action" (Ob) conditions was fully symmetric and the order in which participants were instructed about them was counterbalanced across participants. We described the "observation" (Cb) condition as follows: "The computer is drawing cards from one of the two decks. The goal of the game is to identify the deck from which cards are drawn. The computer changes deck from time to time, without warning." Each "observation" block was preceded by an instruction screen, indicating the association between keys and decks for the block (left key for the orange deck and right key for the blue deck, or vice versa) counterbalanced across blocks. We described the "action" (Ob) condition as follows: "You are drawing cards from one of the two decks by pressing a key. The goal of the game is to draw cards only from the target deck, by identifying the key it is associated with. The association between keys and decks reverses from time to time, without warning." Each "action" block was preceded by an instruction screen, indicating the target deck for the block (orange or blue) counterbalanced across blocks. Each block was initiated by a response (i.e., a random guess) before the onset of the first stimulus sequence. As can be seen from the descriptions of the two conditions, participants were instructed explicitly about

the presence of reversals, but not about their exact frequency (hazard rate $h$). Reversals defined "episodes" during which the hidden state $s$ (the drawn deck in the "observation" condition or the target-drawing key in the "action" condition) is fixed. The length of these hidden-state episodes was sampled pseudo-randomly from a truncated exponential probability distribution (minimum: 4 trials, maximum: 24 trials) to achieve an approximately constant hazard rate $h$ over the course of each block.

The main task (divided in 8 blocks of 72 trials, 4 of each condition) was preceded by a short practice period (2 blocks of 54 trials, 1 of each condition) in which the concentration $\kappa$ of stimulus distributions was increased to 1.5 (instead of 0.5 in the main task) such that participants could understand the structure of the task (in particular, the presence of reversals in the two conditions) with lower uncertainty regarding the generative category $c$ of each stimulus sequence. Performance in the practice blocks was also used to assess participants' understanding of the two conditions. All tested participants were able to perform both conditions at near-ceiling performance after 54 trials of practice.

Each condition ($n = 4$ blocks) consisted of 2 types of blocks as follows: "more stable" blocks ($n = 2$) that contained 6 hidden-state episodes (i.e., a hazard rate $h$ of 1/12) and "more volatile" blocks that contained 12 hidden-state episodes (i.e., a hazard rate $h$ of 1/6). The shorter practice blocks (54 trials instead of 72) contained 6 hidden-state episodes (i.e., a hazard rate $h$ of 1/9, in between the values used in the more stable and more volatile blocks). The eight blocks of the main task were organized in pairs of blocks of the "observation" and "action" conditions, one of each volatility, whose order was counterbalanced both within and between participants. The counter-balancing of the different aspects of the task across participants required to test a multiple of $n = 8$ participants. Given the absence of prior effect sizes for the difference between "observation" (Cb) and "action" (Ob) conditions, we chose a sample size ($n = 24$), which exceeded the average sample size used in human MEG studies at the time of data collection. All results presented in the main text collapse across the two types of blocks within each condition, unless noted otherwise.

**Magnetoencephalography**. MEG data were recorded using a whole-head Elekta Neuromag TRIUX system (Elekta Instrument AB, Stockholm, Sweden) composed of 204 planar gradiometers and 102 magnetometers, at a sampling frequency of 1000 Hz. Prior to the experiment, each participant's head shape was digitized in the MEG coordinate frame and four additional head position indicator (HPI) coils—whose positions were also digitized—were used to monitor and correct for small head movements across blocks. The movement of HPI coils between blocks remained small (mean: 2.3 mm) and did not differ when comparing blocks of the same condition and blocks of different conditions ($t_{23} = 1.6$, $p = 0.128$).

As the first preprocessing step, magnetic noise from external sources was removed using temporal Signal Space Separation, after removing manually detected non-physiological jumps in MEG signals. Stereotyped ocular and cardiac artifacts were corrected using a supervised principal component analysis (PCA) procedure. First, the onset of artifacts (either eye blinks or cardiac R peaks) was detected automatically on auxiliary electrodes (electrooculogram and electrocardiogram) synchronized with MEG signals using a threshold-based approach. MEG signals were then epoched from 200 ms before to 800 ms after artifact onsets and a PCA was used to extract the spatial components of cardiac and ocular artifacts. Typically, one stereotyped PCA component was removed from continuous MEG signals for eye blinks, and two components for heart beats. Continuous MEG signals were high-pass filtered at 0.5 Hz, down-sampled to 500 Hz, and epoched from 200 ms before to 800 ms after the onset of each stimulus ($n = 1440$ per participant and per condition). Finally, epoched MEG signals were low-pass filtered at 8 Hz and their analytical representations (decompositions into real and imaginary parts) were computed using the Hilbert transform, resulting in twice the number of MEG signals (408 from planar gradiometers and 204 from magnetometers). This preprocessing pipeline was implemented using the FieldTrip[46] toolbox (http://www.fieldtriptoolbox.org) and additional custom scripts written in MATLAB.

**Bayes-optimal inference**. Let $L_t$ denote the posterior log-odds belief in the current hidden state $s_t$ at trial $t$, after observing stimulus sequences up to trial $t$ denoted $\Theta_{1:t}$:

$$L_t \equiv \log\left(\frac{p(s_t = 1|\Theta_{1:t})}{p(s_t = 2|\Theta_{1:t})}\right) \tag{1}$$

Given the hazard rate $h$, as previously derived[15], the prior log-odds belief at trial $t + 1$ corresponds to the following expression:

$$\mathcal{F}(L_t) \equiv L_t + \log\left(\frac{1-h}{h} + e^{-L_t}\right) - \log\left(\frac{1-h}{h} + e^{+L_t}\right) \tag{2}$$

Upon observing the new stimulus sequence $\Theta_{t+1} \equiv \{\theta_1, \ldots, \theta_n\}$, the belief is updated using Bayes' rule by combining the prior log-odds belief $\mathcal{F}(L_t)$ with the log-odds evidence $\mathcal{L}_{t+1} \equiv \{\ell_1, \ldots, \ell_n\}$ provided by the new stimulus sequence, corresponding to:

$$L_{t+1} = \mathcal{F}(L_t) + \mathcal{L}_{t+1} \tag{3}$$

As derived in a previous work[17], the log-odds evidence $\mathcal{L}_{t+1}$ provided by the new stimulus sequence is defined as:

$$\mathcal{L}_{t+1} \equiv \sum_{i=1}^{n} \log\left(\frac{p(\theta_i|s_{t+1}=1)}{p(\theta_i|s_{t+1}=2)}\right) \tag{4}$$

$$\ell_{t+1} = \sum_{i=1}^{n} \kappa\left(\cos\left(2(\theta_i - \mu(s_{t+1}=1))\right) - \cos\left(2(\theta_i - \mu(s_{t+1}=2))\right)\right) \tag{5}$$

$$\ell_{t+1} = \sum_{i=1}^{n} 2\kappa\cos\left(2(\theta_i - \mu(s_{t+1}=1))\right) \tag{6}$$

where $\mu(s_{t+1}=1)$ indicates the generative mean of the stimulus sequence $\Theta_{t+1}$, provided that the current state $s_{t+1} = 1$. In the Cb condition, $(s_{t+1}=1) \equiv (c_{t+1}=A)$. Therefore:

$$\mathcal{L}_{t+1} = \sum_{i=1}^{n} 2\kappa\cos\left(2(\theta_i - \mu_A)\right) \tag{7}$$

In the Ob condition, $(s_{t+1}=1) \equiv \{L \to (c_{t+1}=A), R \to (c_{t+1}=B)\}$. Therefore:

$$\mathcal{L}_{t+1} = \begin{cases} \sum_{i=1}^{n} 2\kappa\cos\left(2(\theta_i - \mu_A)\right) & \text{if } r_t = \text{L}, \\ \sum_{i=1}^{n} 2\kappa\cos\left(2(\theta_i - \mu_B)\right) & \text{if } r_t = \text{R}. \end{cases} \tag{8}$$

In the Cb condition, optimal decision-making at trial $t$ corresponds to choosing according to the sign of the posterior log-odds belief $L_t$. In the Ob condition, selecting the action that draws sequences from a target category at trial $t$ corresponds to choosing according to the sign of the prior log-odds belief $\mathcal{F}(L_t)$ carried over at trial $t + 1$. However, by the above expression of $\mathcal{F}(L_t)$, we can guarantee that $\mathcal{F}(L_t)>0$ as long as $L_t>0$ and $h<1/2$, such that it is formally equivalent to cast choices based either on the value of $L_t$ or $\mathcal{F}(L_t)$. Thus, we can use the same Bayes-optimal process in the Cb and Ob conditions.

The fact that the two conditions can be modeled by the same Bayes-optimal process does not guarantee that the two conditions are perfectly matched in terms of provided evidence for any given pair of blocks. Indeed, as can be seen in the above equations, the log-odds evidence $\mathcal{L}_t$ depends on the previous response $r_{t-1}$ in the Ob condition, whereas it does not in the Cb condition. To achieve a perfect match between conditions, we equalized the evidence $\mathcal{L}_{1:T}$ provided by stimulus sequences, where $T$ is the total number of trials per block ($T = 72$), using the following procedure. All orientations $\theta_i$ appear in the above equations as tilts from a category mean $\mu$. We thus pre-generated each stimulus sequence not as orientations $\theta_i$, but as tilts $\delta_i$, from an arbitrary mean. We then defined $\theta_i \equiv \text{mod}(\delta_i + \mu_t, \pi)$, where $\mu_t$ is the category mean at trial $t$. In the Ob condition, if $s_t = 1$, $\mu_t = \mu_A$. Therefore:

$$\mathcal{L}_t = \sum_{i=1}^{n} +2\kappa\cos(2\delta_i) \tag{9}$$

In addition, if $s_t = 2$, $\mu_t = \mu_B$. Therefore:

$$\mathcal{L}_t = \sum_{i=1}^{n} -2\kappa\cos(2\delta_i) \tag{10}$$

In the Ob condition, if $s_t = 1$,

$$\mu_t = \begin{cases} \mu_A & \text{if } r_{t-1} = \text{L}, \\ \mu_B & \text{if } r_{t-1} = \text{R}. \end{cases} \tag{11}$$

Therefore, we obtain the same expression for the log-odds evidence $\mathcal{L}_t$ as in the Cb condition, now irrespective of the previous response $r_{t-1}$:

$$\mathcal{L}_t = \sum_{i=1}^{n} +2\kappa\cos(2\delta_i) \tag{12}$$

Further, if $s_t = 2$,

$$\mu_t = \begin{cases} \mu_B & \text{if } r_{t-1} = \text{L}, \\ \mu_A & \text{if } r_{t-1} = \text{R}. \end{cases} \tag{13}$$

Therefore, we obtain again the same expression as in the Cb condition:

$$\mathcal{L}_t = \sum_{i=1}^{n} -2\kappa\cos(2\delta_i) \tag{14}$$

Thus, by using the same pre-generated tilts $\delta_i$ for a Cb block and an Ob block, we can perfectly equalize the evidence $\mathcal{L}_{1:T}$ provided by all stimulus sequences, resulting in the exact same trajectory of beliefs across conditions. This is because—and despite the dependence of stimulus orientations $\Theta_t$ on the previous response $r_{t-1}$, which is constitutive of the Ob condition—the evidence $\mathcal{L}_t$ does not depend on $r_{t-1}$ but only on the current hidden state $s_t$, whose temporal evolution over the course of a block can also be fully matched between conditions.

**Noisy Bayesian inference**. We assume the possible presence of internal variability, or "noise," at two distinct points of the decision-making process[17]. First, inference noise $\sigma_{\text{inf}}$ occurring during the processing of each stimulus sequence—i.e., the updating of beliefs based on the evidence provided by the stimulus sequence (referred to as $\sigma$ in the main text). Second, selection noise $\sigma_{\text{sel}}$ during response selection, equivalent to a "softmax" choice policy.

Inference noise reflects additive internal variability in the sequential updating of beliefs based on the evidence provided by each stimulus. Updating the current belief based on $n$ pieces of log-odds evidence introduces $n$ pieces of noise. If each of these pieces of noise is i.i.d. and normally distributed with zero mean and variance $\sigma_{\text{inf}}^2$, then the noise introduced by the processing of the sequence has variance $n\sigma_{\text{inf}}^2$. As described in previous work[3], inference noise turns the deterministic belief

update equation into a stochastic draw:

$$L_t \sim \mathcal{N}\left(\mathcal{F}(L_{t-1}) + \mathcal{L}_t, n\sigma_{\text{inf}}^2\right) \tag{15}$$

Selection noise does not perturb the inference process itself and only corrupts response selection. We model such internal variability by sampling responses from the sign of a normally distributed decision variable with mean $L_t$ and variance $\sigma_{\text{sel}}^2$. In some analyses, we also considered the presence of response lapses—i.e., trials in which the participant blindly repeats the previous response instead of selecting the response based on the posterior belief. This leads to a fraction $p_{\text{lapse}}$ of trials associated with a probability of repeating the previous response $p(r_t = r_{t-1}) = 1$.

**Model-based analysis of behavior**. For a block of $T$ trials, each fully specified by its stimulus sequence $\Theta_t$, we observed a series of responses $r_1, \ldots, r_T$. We fitted the parameter values $\phi \equiv \left\{h, \sigma_{\text{inf}}, \sigma_{\text{sel}}, p_{\text{lapse}}\right\}$, which resulted in the best match between the observed series of responses and the series of responses predicted by the model, $p(r_{1:T}|\Theta_{1:T}, \phi)$. We also fitted "reduced" models that removed certain components by fixing their corresponding parameters to zero. For example, a model without selection noise would correspond to $\sigma_{\text{sel}} = 0$.

We found the best-fitting parameter values by sampling from the Bayesian posterior over parameters using particle MCMC methods[47]. These methods use standard MCMC methods to sample from the parameter posterior, but replace computation of the parameter likelihood $p(r_{1:T}|\Theta_{1:T}, \phi)$—not possible in closed form—with a noisy but unbiased approximation of this likelihood by a particle filter. As the MCMC method, we used the adaptive mixture Metropolis method[48] that adapts its proposal distribution in an initial burn-in period to achieve favorable acceptance ratios. Prior distributions over parameters $p(\phi)$ were defined as truncated normal distributions ($h$: mean = 0.2, SD = 0.1, range = [0,1]; $\sigma_{\text{inf}}$: mean = 0.5, SD = 0.2, range = [0,10]; $\sigma_{\text{sel}}$: mean = 0.5, SD = 0.2, range = [0,10]; $p_{\text{lapse}}$: mean = 0.01, SD = 0.05, range = [0,1]). It is noteworthy that the particle filter also provided an unbiased estimate of the belief trajectory $p(L_{1:t}|r_{1:t}, \Theta_{1:t}, \phi)$ for $t \in [1, T]$, which was used in subsequent analyses of MEG signals. The particle MCMC fitting procedure was written in Julia v1.0[49] (https://julialang.org) and was run using 10,000 particles.

We used the unbiased estimate of the marginal likelihood $p(r_{1:T}|\Theta_{1:T})$ computed using conditional likelihoods from the particle filter as model evidence in BMS analyses. This metric integrates over parameters $\phi$ and thus penalizes model complexity[50] without requiring explicit penalization terms as in the Bayesian Information Criterion. BMS was conducted using separate fixed-effects and random-effects approaches, which yielded qualitatively identical results. The fixed-effects approach assumes that all participants are relying on the same model and consists in comparing the log-marginal likelihood summed across participants for each tested model. By contrast, the random-effects approach assumes that different participants may rely on different models and consists in estimating the distribution over models that participants draw from ref. [51]. We used the Dirichlet parameterization of the random-effects approach implemented in SPM12 (Wellcome Center for Human Neuroimaging; http://www.fil.ion.ucl.ac.uk/spm).

**Psychometric analysis of behavior**. Reversal learning behavior was characterized by two psychometric curves. First, we fitted response reversal curves—i.e., the probability of correctly identifying the hidden state of the current episode as a function of trial position in the current episode—by a saturating exponential function:

$$p(r_k = s) = p_0 + (p_{\text{rev}} - p_0)\left(1 - e^{-\frac{k}{t_{\text{rev}}}}\right) \tag{16}$$

where $p(r_k = s)$ corresponds to the probability of correctly identifying the current hidden state $s$ at trial $k$, with $k$ the position of the trial in the current episode. The two fitted parameters are the reversal time constant $t_{\text{rev}}$ and the asymptotic reversal rate $p_{\text{rev}}$. The initial reversal rate $p_0$ is set to 0.5 for the first episode of each block and to $1 - p(r_k = s)$ reached at the last trial of the previous episode for subsequent episodes. Best-fitting parameter values for $t_{\text{rev}}$ and $p_{\text{rev}}$ were obtained by maximum likelihood estimation (MLE) through gradient descent (on the negative log-likelihood) using the interior point algorithm implemented in MATLAB.

Second, we fitted response repetition curves—i.e., the probability of repeating the previous response as a function of the evidence provided by the intervening sequence in favor of the previous response—by a three-parameter sigmoid function:

$$p(r_t = r_{t-1}) = p_{\text{rep}} + (1 - p_{\text{rep}})\frac{1}{1 + e^{-\beta\mathcal{L}_{\text{rep},t} + \beta_{\text{rep}}}} \tag{17}$$

where $p(r_t = r_{t-1})$ corresponds to the probability of repeating the previous response and $\mathcal{L}_{\text{rep},t}$ corresponds to the log-odds evidence provided by the stimulus sequence at trial $t$ in favor of the previous response $r_{t-1}$:

$$\mathcal{L}_{\text{rep},t} \equiv \sum_{i=1}^{n} \log\left(\frac{p(\theta_i|s_t = r_{t-1})}{p(\theta_i|s_t \neq r_{t-1})}\right) \tag{18}$$

The three fitted parameters are the slope of the sigmoid curve $\beta$ (indexing the sensitivity of responses to the evidence $\mathcal{L}_{\text{rep},t}$), the decision criterion for repeating the previous response $\beta_{\text{rep}}$ and the lower asymptote of the sigmoid curve $p_{\text{rep}}$

(reflecting the fraction of evidence-independent repetitions). As above, the best-fitting parameter values were obtained by MLE. The decision criterion $\beta_{rep}$ was reparametrized as the PSE—i.e., the amount of evidence $\mathcal{L}_{rep,t}$ for which $p(r_t = r_{t-1}) = 0.5$.

**Multivariate pattern analyses of MEG signals.** We applied multivariate pattern analyses to stimulus-locked MEG epochs, to estimate the neural patterns associated with four characteristics of each stimulus $i$, where $i$ corresponds to the position of the stimulus in question in the current sequence: (1) its orientation $\theta_i$ described by $\cos(2\theta_i)$ and $\sin(2\theta_i)$; (2) its change (tilt) from the previous stimulus, described by $|\theta_i - \theta_{i-1}|$; (3) the strength of the evidence provided by the stimulus, described by its tilt from the nearest category boundary $\theta_0$, $|\theta_i - \theta_0|$; and (4) the consistency of the stimulus with the current belief $L_{t,i}$, described as its tilt from the nearest category boundary signed by the direction of the provided evidence $\ell_i$ in relation to $L_{t,i}$. Importantly, owing to features of our experimental task (in particular, the changes in category means $\mu_A$ and $\mu_B$ across trials), these four stimulus characteristics showed no significant correlation with each other for a given stimulus $i$ nor between successive stimuli $i$ and $i + 1$ (all $r^2 < 0.001$).

We used a cross-validated multivariate linear encoding model to estimate the spatial MEG patterns $\hat{w}$ associated for each participant with each stimulus characteristic $x$ at every time point from 100 ms before to 700 ms after stimulus onset. For each cross-validation fold ($n = 12$, interleaved), we defined spatial MEG patterns $\hat{w}$ on the training set by regressing—in a least-squares sense—each $z$-scored gradiometer feature $\mathcal{Z}_{train}$ (real and imaginary parts, yielding 408 features in total) against the stimulus characteristic $x_{train}$ across stimulus exemplars ($n = 1320$ for each cross-validation fold), by solving $\text{argmin}_w(\mathcal{Z}_{train} - wx_{train}^2)$ for each MEG feature. We then projected the MEG data on the test set $\mathcal{Z}_{test}$, on the dimension defined by the coding weights $\hat{w}$ to obtain neural predictions the stimulus characteristic $x_{test}$ for each epoch of the test set ($n = 120$ for each cross-validation fold). After applying this procedure for each cross-validation fold, we computed the linear correlation coefficient between neural predictions $\hat{x}$ and ground-truth values $x$ of the stimulus characteristic. The coding precision metric reported in the main text corresponds to the Fisher transform of the correlation coefficient, which is approximately normally distributed, such that we could compute standard parametric statistics at the group level.

The multivariate pattern analyses described above were conducted for each participant and each condition. At the group level, we used standard parametric tests (paired $t$-tests, repeated-measures analyses of variance (ANOVAs)) to assess the statistical significance of observed differences in coding precision between conditions across tested participants. Neural coding latency was computed for each stimulus characteristic and each condition by estimating the peak of coding precision using a jackknifing (leave-one-out) procedure[52]. The type 1 error rate arising from multiple comparisons was controlled for using non-parametric cluster-level statistics computed across time points[53]. All findings reported in the main text were robust to changes in the method used for computing spatial MEG patterns (e.g., by applying ridge-regression decoding instead of least-squares encoding) and to the number of cross-validation folds.

We accounted for the unbalance in trial number between sequences leading to response repetitions and response reversals when decoding consistency at each stimulus position (Fig. 6d). As the smaller number of sequences leading to response reversals could artificially reduce the coding precision of consistency, we applied the same spatial MEG patterns for the two types of sequences ("common-filters" approach). These spatial patterns (coding weights) were estimated through cross-validation based on sequences leading to response repetitions, separately for the Cb and Ob conditions. For similar reasons, when assessing the effect of the current belief magnitude on the neural coding of stimulus characteristics (Fig. 7c), we applied the same spatial MEG patterns for each bin. These spatial patterns were estimated through cross-validation across all bins, separately for the Cb and Ob conditions.

**Source reconstruction of MEG signals.** Cortical surface-based segmentation and reconstruction was performed with the FreeSurfer image analysis suite (http://surfer.nmr.mgh.harvard.edu) using the default surface-based pipeline[54,55], based on high-resolution T1-weighted anatomical magnetic resonance images recorded from each participant using a Verio 3T scanner (Siemens AG, Munich, Germany) and a 64-channel head coil, driven by a three-dimensional magnetization-prepared rapid acquisition with gradient echo sequence, with an isotropic resolution of 1 $mm^3$. Obtained anatomical surfaces were down-sampled to 5000 vertices and realigned to the MEG spatial coordinate frame using each participant's digitized head shape and fiducial markers. Source reconstruction was performed with Brainstorm[56] (http://neuroimage.usc.edu/brainstorm) using weighted minimum norm estimation with Tikhonov regularization[57]. Forward modeling was performed using overlapping spheres. We obtained noise covariance matrices from 3 min of "empty room" recordings[58] on which we applied the exact same pre-processing pipeline, using a diagonal regularization $\lambda$ of 0.1. Vertex-wise inverse operators were computed based on MEG signals from both gradiometers and magnetometers (non-Hilbert transformed) by constraining dipole orientations to be orthogonal to the cortical surface.

Source reconstruction was conducted for each participant based on his/her individual cortical surface. The results of analyses performed in source space were then co-registered to a common anatomy (the ICM152 MNI template) for group-level statistics based on the FreeSurfer registration procedure. In practice, vertex-level results obtained for each participant were interpolated from the subject-specific anatomy to the common anatomy using inverse distance weighting as implemented in Brainstorm (Shepard's method).

**Searchlight analysis of MEG signals.** We conducted "'searchlight"-based multi-variate pattern analyses on the activity of reconstructed cortical sources (vertices) from each participant. After decomposing the activity of each vertex into its real and imaginary parts using the Hilbert transform (as done for sensor-level analyses), we computed coding precision at each vertex position and each time point around stimulus onset by using as features the current vertex and its ten closest neighbors in terms of Euclidean distance. To increase the spatial selectivity of searchlight-based analyses, we computed for each stimulus characteristic and each vertex position a selectivity metric, corresponding to the result of a jackknifed $F$-test, which compares coding precision of this characteristic to the coding precision of other characteristics. We accounted for overall differences in coding precision between characteristics by normalizing the coding precision of each characteristic at each vertex position by the peak coding precision of this characteristic across all time points and vertices. Non-parametric cluster-level statistics[53] were computed across time points and vertices as implemented in FieldTrip[46]. We marginalized these cluster-level statistics across either space or time by summing statistics across the corresponding dimension for display purposes.

**Cross-condition generalization analysis of MEG signals.** We conducted cross-condition generalization analyses of spatial MEG patterns to assess the degree of similarity between neural representations of stimulus characteristics across conditions. In practice, we used the coding weights estimated on the training set of one condition to compute neural predictions on the test set of the other condition. Although these cross-condition generalization analyses do not require cross-validation to avoid overfitting, as the two conditions are non-overlapping, we wanted to compare coding precision scores between within-condition (non-generalized) and between-condition (generalized) settings. We thus applied the same 12-fold cross-validation approach within each condition (which had the same number of stimulus exemplars) such that: (1) coding weights $\hat{w}$ were estimated using the same amount of training data in within- and between-condition settings, and (2) coding precision was computed using the same amount of test data in the two settings.

To quantify the degree of similarity between two neural codes based on the results of cross-condition generalization analyses, we simulated and fitted a fixed number $n$ of linear coding units $z_i$ ($n = 300$) to within- and between-condition coding precision scores obtained from the MEG data. A fixed number $n_{sel}$ of these units ($n_{sel} = 100$) was defined as "selective" to a stimulus characteristic $x$, meaning that their activity scales linearly with $x$ (Fig. 5a):

$$z_i = \beta_i x + \varepsilon \quad (19)$$

where $\beta_i = 1$ for selective units and 0 for non-selective units, and $\varepsilon \sim \mathcal{N}(0, \sigma)$ corresponds to background noise of variance $\sigma^2$ (i.i.d. across units). We used the same cross-validated multivariate linear encoding model and the same amount of data, to estimate coding patterns $\hat{w}$ for the stimulus characteristic $x$ from population activity and obtain predictions $\hat{x} \equiv \hat{w}^T Z$. Simulating the same population of units in two conditions was controlled by three parameters: the amount $\sigma$ of background noise in each condition ($\sigma_{Cb}$ and $\sigma_{Ob}$) and the fraction of units $\omega$ selective to the stimulus characteristic $x$ in both conditions (coding similarity). Parameters $\sigma_{Cb}$ and $\sigma_{Ob}$ scaled negatively with coding precision within each condition (values 1–2 in Fig. 5), whereas parameter $\omega$ scaled positively with coding precision between conditions (values 3–4 in Fig. 5). We fitted all three parameters to the four coding precision values obtained from the cross-condition generalization analysis, using a jackknifed least-squared error minimization procedure. Importantly, the best-fitting estimate of coding similarity $\omega$ was robust to changes in the total number $n$ of units and to the number $n_{sel}$ of selective units (both fixed across conditions).

**Estimation of neural hazard rate from MEG signals.** We used the estimate of the belief trajectory $p(L_{1:t}|r_{1:t}, \Theta_{1:t}, \phi)$ for $t \in [1, T]$ provided by the particle filter—conditioned on each participant's responses $r_{1:t}$, stimulus sequences $\Theta_{1:t}$, and model parameters $\phi$ to fit the "neural" hazard rate $h^*$ for which we could best decode consistency from MEG signals. For this purpose, we computed belief trajectories for varying values of the perceived hazard rate $h$ (Fig. 6a)—i.e., ignoring the best-fitting value obtained from behavior—and derived for each of these trajectories the consistency of each stimulus in relation to the estimated belief conditioned on each value of $h$. These alternative instantiations of the same consistency variable $x_{cons}$ were used to compute "tuning curves" of coding precision as a function of $h$. We then fitted the hazard rate value $h^* \equiv \text{arg max}_h(\text{corr}(\hat{x}_{cons}, x_{cons}))$ for each participant and condition, where $x_{cons}$ depends implicitly on $h$ and $\hat{x}_{cons}$ corresponds to neural predictions obtained using the multivariate pattern analysis described above. To obtain smooth estimates of $h^*$,

we approximated each tuning curve by a quadratic polynomial function using a least-squared error minimization procedure and computed its maximum. Importantly, using unsmoothed estimates of $h^*$ did not change the pattern of findings.

**Statistical testing.** Unless noted otherwise, statistical analyses of differences between scalar metrics across experimental conditions relied on two-tailed parametric tests (paired $t$-tests, repeated-measures ANOVA) across tested participants in a paired (within-subject) manner. Given our sample sizes, these statistical tests were applied outside the small-sample regime, thereby matching their basic assumptions.

**Reporting summary.** Further information on research design is available in the Nature Research Reporting Summary linked to this article.

## Data availability

The manuscript includes all datasets generated or analyzed during this study. The behavioral data from the main magnetoencephalography experiment and the control behavioral experiment are publicly available at https://doi.org/10.6084/m9.figshare.13200128. The magnetoencephalography data are available from the corresponding authors upon request. Source data are provided with this paper.

## Code availability

The analysis code supporting the reported findings are available from the corresponding authors upon request.

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

## Acknowledgements

This work was supported by a starting grant from the European Research Council (ERC-StG-759341) awarded to V.W., a junior researcher grant from the French National Research Agency (ANR-14-CE13-0028-01) awarded to V.W., a France-US collaborative research grant from the French National Research Agency (ANR-17-NEUC-0001-02) and the National Institute of Mental Health (1R01MH115554-01) awarded to V.W. and J.D., and a department-wide grant from the Agence Nationale de la Recherche (ANR-17-EURE-0017). A.W. was supported by the FIRE Doctoral School. V.C. was supported by the French National Research Agency (ANR-16-CE37-0012-01). J.D. was supported by the James S. McDonnell Foundation (grant #220020462).

## Author contributions

V.W. conceptualized and supervised the project. A.W., V.C., J.K.L., and V.W. conducted the experiments and pre-processed the data. J.D. and V.W. developed the computational model. A.W. and V.W. conducted the data analyses. A.W., V.C., and V.W. wrote the original manuscript. A.W., V.C., J.K.L., J.D., and V.W. reviewed and edited the manuscript. J.D. and V.W. acquired funding.

## Competing interests

The authors declare no competing interests.
