## [Peer Review File · Nature Communications]

Reviewers' comments:

Reviewer #1 (Remarks to the Author):

This study compares two modes of hidden-state inference in changing environments: one based on only observing sequences of ambiguous sensory cues ("cue-based" condition); the other based on the outcomes (again presented as sequences of sensory cues) of the agent's own choices ("outcome-based" condition). Both conditions are matched exactly in terms of sensory evidence, and can be solved by the exact same Bayes-optimal algorithm.

The authors find stronger tendency to repeat the previous choice in the outcome-based condition. Using an established normative model for this situation (Glaze et al, 2015), this behavioral difference can be attributed to a lower subjective hazard rate. The results are overall consistent with the idea that participants perceive the environment to be more stable in the outcome-based condition, which the above model translates into stronger prior beliefs. This effect produces a double-dissociation between two block types that differing in objective hazard rate (stable vs. volatile): better performance for the outcome-based condition in stable blocks and worse performance for the outcome-based condition in volatile blocks.

The above behavioral differences are mirrored by an effect of the consistency between prior and current evidence on the pattern of MEG responses evoked by individual cues, an effect that is localized to medial temporal lobes.

The work has several strengths - first and foremost, the elegant task design and behavioral modeling. The main finding, the difference in perceived environmental stability (and prior strength) between the cue-based and outcome-based condition is quite interesting. That said, it should be clarified why this result matters (see below).

I find the MEG results to be less informative about the issue at stake, despite the fact that the analyses are clever and deep. The authors dissociate the decoding of sensory input (cue orientation), sensory evidence (orientation relative to category boundary), and the orientation change from cue to cue -- three quantities which are elegantly orthogonalized in the design, although I do have an issue with one aspect of the approach (see point 3 below). They also present a compelling case for shared neural representations of these three quantities between the two tasks. Yet, studying the neural representation of stimulus features and decision evidence is not the key point of this study. And the functional relevance of the neural "consistency signal" is unclear (see below). It seems that this is simply an indirect reflection of the main behavioral result: the difference in subjective hazard rate, and hence, prior strength.

Overall, I suggest focusing this manuscript on the behavioral results and use the MEG data mainly to establish that the neural encoding of both sensory input, and decision evidence, is also matched between both tasks. In what follows, I list a number of specific points that should be addressed in revision.

Major:

1. Explain conceptual significance.

The tasks are very carefully designed, rendering the comparison highly specific. But the authors should do a better job in explaining why it actually addresses an important issue: Why does it matter that perceived environmental stability depends on the interaction with the environment? The implications of the findings should be explained in Discussion.

2. Confounding factors.

Apart from control over information sampling, both tasks differ along another dimension: the cue-based condition entails judging a *previous* evidence sequence; the outcome-based conditions entails drawing the category for the *subsequent* evidence sequence. Can the authors rule out that this pre vs. post difference, rather than control over sampling, is the decisive factor accounting for the observed difference in perceived stability? If not, what would this mean for the broader implications? This issue needs to be discussed.

3. Rationale for MEG searchlight decoding analysis.

The searchlight decoding analysis is carried out in a way that shows only regions, in which a single variable can be decoded better than others. In other words, regions which exhibit similar decoding precision of multiple variables get under the radar.

The theoretical justification of this choice is unclear. It seems to be driven by a general desire to present a picture of neat regional specialization (e.g., p. 17, 2nd par: "To increase the spatial selectivity of obtained results..."). Cortical regions commonly contain mixtures of neurons selective for multiple different variables, not a single one. Indeed, even single neurons commonly encode multiple variables. That does not mean that these regions / neurons are irrelevant for the transformation of any particular variable. The authors should show all regions that exhibit significant decoding of any variable, irrespective of decoding of other variables.

Note that the above approach not only misses an important aspect of cortical function, it also limits the of conclusions that can be drawn from the results. Specifically, I don't think that the conclusion (p. 17, same par.) "Together, these results highlight a spatiotemporal cascade of neural patterns involved in the processing of each stimulus, ..." is valid; supporting this conclusion would require showing full maps and time courses of decoding for each variable, irrespective of the others.

4. Neural "consistency signal".

a. Much of the discussion focusses on the MTL result. Apart from my doubt about the general approach (see point 3), it is unclear why a neural signal measuring consistency between current belief and new evidence and matters for the process under study: this quantity plays no role in the decision computation described on pages 40-41.

b. The progression from Fig. 5 to 6, and the associated text, are also confusing. In light of the dynamic belief updating model used here, the obvious way of quantifying consistency is by means of the "belief trajectory analysis" from Fig. 6. This uses the instantaneous (transformed) belief state before each new

cue as prior, against which consistency of the evidence is assessed. The analysis from Fig. 5 uses only the prior at the start of trial, thus ignoring possible changes of belief state during the course of the trial. There is a clear conceptual advantage for the former analysis (Fig. 6) over the latter (Fig. 5) – Fig. 5 thus seems like an unnecessary detour. It's not clear what this (suboptimal) analysis adds to the story. I suggest dropping this altogether.

Minor:

- The first paragraph of Introduction is confusing and misleading. It refers to a review by Shadlen & Shohamy that compares the information sources for perceptual and value-based decisions: sensory environment vs. memory. The authors move on to claim that the key differences between perceptual and value-based decisions lies in the degree of control over information sampling. This presentation conflates two separate dimensions along which perceptual and value-based decisions differ (information source and control over sampling). It also prompts the reader to expect a direct comparison between the impact of these two dimensions, which this study clearly does not deliver. The paragraph should be re-written.

- The data and model predictions look compelling; yet the authors should present more evidence that model fits the data well.

- How could the repetition effects in Fig. 2 be produced by any parameter in the model used, other than subjective hazard rate? The claim that this result "offers a computational account for the increased PSE in the outcome-based condition" seems overstated.

- Are all results except Fig. 3e collapsing across the two objective hazard rates? This needs to be explicitly stated.

- The objective hazard rates should be plotted along with the subjective ones in Fig. 3b, for reference. I realized that there are two h values intermixed; perhaps then present the pooled h – after all, the behavioral modeling also seems to pool across these two contexts.

Reviewer #2 (Remarks to the Author):

In this paper the authors examine how people detect changes in the state of the environment during a reversal learning task. In this task sensory stimuli are drawn from one out of two categories with the generative category changing infrequently and at random points. Unlike previous studies, here the authors compare behaviour in two normatively equivalent conditions. In the first condition ("cue-based"), participants first observe and then categorise the sensory stimulus. In the second condition ("outcome-based") participants make a prediction about the categorical identity of the stimulus in the forthcoming trial. The authors report that in the outcome-based condition people are more prone to miss switches in the generative state of the environment, which translates into a larger choice repetition

rate. These results are attributed to a stronger reliance on prior information, which in turn falls out from an overestimation of the stability of the environment. Further analyses using magnetoencephalography (MEG) confirm and further corroborate the behavioural results.

Overall, this is a very well-designed experiment addressing the important question of belief updating in relation to the way people interact with information. The reported behavioural effect appears to be robust, however an explanation of this effect is currently missing. Why do people rely on prior information more strongly in the outcome-based condition and does this necessarily imply a larger sense of stability? Below I outline the points of criticism in detail.

- 1) There is an overall larger choice repetition rate in the outcome based condition. This effect can be a global effect, equally present in trials of all type, or driven by repetition behaviour in specific trials. The current analysis presented in Figure 2b is a first step towards fully characterising the behavioural differences between the two conditions. However, I recommend that the authors go one step further and perform this analysis by splitting trials in “repetition correct” and “repetition incorrect”. This will shed light on whether participants tend to have different repetition rates when a switch has occurred.
- 2) Participants in the outcome-based condition have an overall smaller perceived hazard rate. Since this is the central finding in the paper, this analysis needs to be further unpacked. First, how does the signed and unsigned difference between the perceived hazard rate and the actual hazard rate differ as a function of the actual hazard rate in the two conditions? Even if people underestimate the hazard rate in the outcome-based condition their estimate could still be closer to the actual hazard rate. Second, how do individual differences look like with regards to the over /under-estimation of hazard rate? For instance, do participants consistently overshoot/ undershoot the hazard rate in both conditions? This more detailed analysis will allow understanding the changes in behaviour between outcome-based and cue-based conditions.
- 3) The finding that participants rely more heavily on the prior belief in the outcome-based conditions is consistent with a lower perceived hazard rate in the normative framework that the authors employ. However, stronger reliance on the prior could just be an idiosyncratic factor independent of perceived stability. In Bayesian terms that would correspond to having a more precise prior distribution in the outcome-based condition. Can this study dissociate the two possibilities? It will also be instructive to consider more naïve Bayesian models in which the hazard rate is not a free parameter.
- 4) In their MEG analysis the authors focus on 3 relevant quantities pertaining to the perceived stimulus (Figure 4). However, it is not clear why they do not consider the neural correlates of the prior belief in a similar fashion. This seems to be an obvious quantity to consider, unless there are confounding factors that prevent the authors from decoding the prior belief (currently not discussed).
- 5) The analysis in Figure 6 appears to just be a coarser version of the analysis presented in Figure 7. I would recommend merging the former with the latter or dropping the former altogether.

Reviewer #3 (Remarks to the Author):

I enjoyed reading this interesting study of probabilistic belief updating with and without control over sampling of information, in outcome-based and cue-based task conditions respectively. The authors found

- * The outcome-based condition had a longer reversal time constant & higher reversal rate asymptote - i.e. more evidence was required to switch response (i.e. a switch 'threshold' was different but response sensitivity the same & perseveration only nonsignificantly higher in outcome condition) as the hazard rate was lower (i.e. the posterior from the last trial contributed more to the prior on next trial).
- * Inference 'noise' was the same across conditions; Bayesian model selection implied no response noise was added.
- * Mean accuracy & RT was no different across cue/outcome conditions, but accuracy higher in outcome condition with few reversals, and lower when more reversals, as one would expect given its low hazard rate
- * Orientation & change in tilt & evidence strength were decoded from the MEG data with the same precision in and across both conditions but...
- * 'Consistency' of evidence with the previous response (i.e. matching/non-matching the prior) was more precisely decoded in the outcome condition - in R MTL (~200 ms) & PPC (~400 ms)...
...probably because of the lower hazard rate (consistency with the current belief was ~equally precise across conditions)
- * Coding of 'relational' evidence (to current beliefs) appeared before 'absolute' evidence & its precision was proportional to magnitude of beliefs, and greater than precision of absolute coding in MTL

The authors have conceived a great question, with a good experiment design, careful and rigorous methodology, and interesting results. The paper itself is very clearly written. Its conclusions will be relevant to numerous fields, and I agree with the authors that they may be especially relevant to psychiatric disorders and OCD in particular. Overall I support its publication, and I have only relatively minor comments:

Parameter recovery looks good, given the correlations across conditions in Fig 3B and 3C. I wondered the same about model recovery: from simulated data with and without a more-or-less noisy response model, does model selection correctly detect the presence or absence of the response model?

P33 - I was a bit puzzled by some of the discussion of the 'relational code'. I might have misunderstood, but to me this interesting analysis is revealing the neural correlates of model-updating in the brain, i.e. the processes that occur when priors and likelihoods interact. But actually describing these correlates as a 'relational code' and saying it is "impractical for downstream read-out" is a bit misleading to me - if the system knows its prior beliefs then is it easy to 'read-out' adjustments to these beliefs.

I was also puzzled by: "Another property of this relational code is that it scales with the magnitude of the current belief: it is almost not expressed when the current belief is weak, but strongly expressed when the current belief is strong – an effect which violates a basic tenet of Bayesian reasoning: the independence between the likelihood and prior terms". Unless I misunderstand, the 'relational code'

reflects the encoding of new evidence given the subject's prior beliefs. In Bayesian belief updating, this new evidence will be a precision-weighted prediction error, and the precision weight will be the ratio of likelihood precision/(likelihood + prior precisions). So one *would* expect the relational code to "scale with the magnitude of [i.e. confidence in] the current belief" and this certainly doesn't violate Bayes (I'm not sure about circular inference).

Another question I have about this conclusion is whether this effect is some artefact of the training process? If there are not many reversals then presumably the classifier is trained on lots of trials when the subject has stronger beliefs, and not so many when the subjects are unsure. So the relational code might scale with the magnitude of the current belief because there are just more trials when subjects are sure? Can you show this isn't the case by training it on balanced numbers of trials when subjects ought to be sure/unsure, perhaps? Or demonstrating that the numbers of such trials are already balanced?

Small points:

P3 - surely the variety of an apple is inferred from both sensory signals and long term memory - the apple's colour could be an example of pure sensory inference? Also is it worth mentioning that in the real world, most cue-based inference is also active (we actively sample information), so really this paper is about active vs passive inference rather than cue vs outcome inference?

P40 - why were the MEG signals low pass filtered at 8 Hz? Doesn't this lose a lot of information?

P44 - why was the prior mean over the hazard rate so high at 0.2? Do you get the same results if you use a lower or empirical prior?

Reviewer #1

This study compares two modes of hidden-state inference in changing environments: one based on only observing sequences of ambiguous sensory cues ("cue-based" condition); the other based on the outcomes (again presented as sequences of sensory cues) of the agent's own choices ("outcome-based" condition). Both conditions are matched exactly in terms of sensory evidence, and can be solved by the exact same Bayes-optimal algorithm.

The authors find stronger tendency to repeat the previous choice in the outcome-based condition. Using an established normative model for this situation (Glaze et al, 2015), this behavioral difference can be attributed to a lower subjective hazard rate. The results are overall consistent with the idea that participants perceive the environment to be more stable in the outcome-based condition, which the above model translates into stronger prior beliefs. This effect produces a double-dissociation between two block types that differing in objective hazard rate (stable vs. volatile): better performance for the outcome-based condition in stable blocks and worse performance for the outcome-based condition in volatile blocks.

The above behavioral differences are mirrored by an effect of the consistency between prior and current evidence on the pattern of MEG responses evoked by individual cues, an effect that is localized to medial temporal lobes.

The work has several strengths - first and foremost, the elegant task design and behavioral modeling. The main finding, the difference in perceived environmental stability (and prior strength) between the cue-based and outcome-based condition is quite interesting. That said, it should be clarified why this result matters (see below).

We thank the reviewer for his/her positive assessment of our work. We hope to have addressed his/her comments using additional quantitative analyses of the behavioral and MEG data and conceptual clarifications in the revised manuscript, but also using an additional behavioral dataset that supports our main finding, that it is the instrumental control conferred to decision-makers that generates the increased perceived environmental stability observed in the outcome-based (Ob) condition. As described below, we have also clarified in the introduction and discussion sections of the revised manuscript why this result should matter to existing theories of learning and decision-making, both in the general population and in specific psychiatric disorders.

I find the MEG results to be less informative about the issue at stake, despite the fact that the analyses are clever and deep. The authors dissociate the decoding of sensory input (cue orientation), sensory evidence (orientation relative to category boundary), and the orientation change from cue to cue -- three quantities which are elegantly orthogonalized in the design, although I do have an issue with one aspect of the approach (see point 3 below). They also present a compelling case for shared neural representations of these three quantities between the two tasks. Yet, studying the neural representation of stimulus features and decision evidence is not the key point of this study. And the functional relevance of the neural "consistency signal" is unclear (see below). It seems that this is simply an indirect reflection of the main behavioral result: the difference in subjective hazard rate, and hence, prior strength.

Overall, I suggest focusing this manuscript on the behavioral results and use the MEG data mainly to establish that the neural encoding of both sensory input, and decision evidence, is also matched between both tasks.

We understand the reviewer's comment. The MEG results did not resonate as much as they could with the main behavioral finding of the manuscript: that interacting with uncertain outcomes (in the outcome-based condition) stabilizes hidden-state inference. However, we do believe that the MEG results provide compelling support in favor of our interpretation of the finding – namely, that interacting with uncertain evidence stabilizes hidden-state inference (rather than increases repetition bias, for example). We have thus modified in depth the manuscript such that the MEG analyses are used much more explicitly to support the main behavioral finding. For this purpose, we have followed every suggestion made by the reviewer (and the other two reviewers), which we think has strengthened the manuscript and its conclusions. Also, we have modified the discussion section to clarify the functional relevance of the consistency signal decoded from MEG signals during hidden-state inference, based on different comments made by the reviewers.

We agree with the reviewer that studying the neural representations of the distinct stages of stimulus processing is not the focus of the study. However, we see this analysis as an important first step to characterize at which stage the difference between the cue-based and outcome-based conditions emerges. Nevertheless, following the reviewer's suggestion, we have tightened the MEG analyses. In particular, we now stress much more explicitly in the results section how the MEG analyses are used to support our interpretation of the main behavioral finding. Note, also, that we now clarify in the results (using a new Supplementary Fig. 10) how the MEG data can answer an important question regarding the temporal scale at which hidden-state inference is performed – and thus the temporal scale at which hidden-state beliefs are updated (Supplementary Fig. 10b). Indeed, the MEG data show that subjects update hidden-state beliefs after each stimulus, not after each sequence when probed for a response (Supplementary Fig. 10c). Behavioral analyses are inherently blind to this distinction, since these two inference schemes result in the same responses. The discussion section features an updated paragraph (p. 21) that describes this dynamic (stimulus-level) inference scheme in light of the recent literature. We hope that the reviewer will find the description of the MEG results improved, and their integration with the behavioral findings clarified. We thank the reviewer from prompting these changes to the manuscript.

In what follows, I list a number of specific points that should be addressed in revision.

Major:

1. Explain conceptual significance.

The tasks are very carefully designed, rendering the comparison highly specific. But the authors should do a better job in explaining why it actually addresses an important issue: Why does it matter that perceived environmental stability depends on the interaction with the environment? The implications of the findings should be explained in Discussion.

We thank the reviewer for this positive assessment of our experimental protocol. We have altered the discussion section to explain earlier and more prominently the implications of our findings. The issue of whether and how interacting with one's environment affects how we learn from it has been

a hugely influential topic in the learning sciences for decades, something which we now stress explicitly in the discussion section (p. 19):

“ A first immediate conclusion is that the status of the decision-maker – an agent interacting with its environment in the outcome-based condition, or an observer contemplating the same environment in the cue-based condition – shapes human learning and decision-making under uncertainty. In the learning sciences, the distinction between ‘active’ and ‘passive’ information gathering has been stressed for decades¹¹. Controlling the flow of incoming information during learning is seen as an important degree of freedom for acquiring non-redundant information and accelerating the learning process. By contrast, and surprisingly, existing theories of cognitive inference either do not consider the degree of control over the sampling of evidence as a contextual variable^{2,4,7,15}, or formulate distinct computational models of ‘active’ and ‘passive’ inference¹⁸. ”

We now also explain earlier in the discussion section (pp. 19-20) why and how the presence of control may trigger the observed stabilization of hidden-state inference. By discussing the possible driving forces of the observed effect, this paragraph offers possible implications for our findings:

“ Interacting with uncertain environments confers several cognitive advantages, which could explain the lower perceived hazard rate and the resulting stabilization of hidden-state inference in the out-come-based condition. First, the active sampling of evidence enables testing current hypotheses in a self-directed, online fashion – an ability known to improve human learning in novel environments^{11,12}. A functional interpretation is that the lower perceived hazard rate in the outcome-based condition aims at testing the current hypothesis that the previous action samples from the target category. Such hypothesis testing is by definition impossible in the cue-based condition because participants had no control over the sampling of evidence. Note that the inference process used in the cue-based condition is not equivalent to ‘observational’ learning – i.e., learning from or about another agent interacting with uncertain evidence^{19,20}. Indeed, participants were actively committing to interpretations regarding the source of evidence in both conditions, the sole difference being the presence or absence of instrumental control over the sampling of evidence. Even without considering hypothesis testing, knowledge about action-outcome contingencies affords an agent to predict incoming sensory signals, and to stabilize these signals by stabilizing its behavior^{18,21,22}. The lower perceived hazard rate in the outcome-based condition may thus aim at stabilizing the source of the incoming evidence – something which is not possible in the absence of control. Last, the difference in perceived hazard rate between the two conditions may betray a temporal hierarchy between two types of changes which can arise in uncertain environments: 1. external (uncontrollable) changes in the state of the environment (as in the cue-based condition), and 2. changes in the instrumental control of the same environment (as in the outcome-based condition). In this view, participants behave as if they assume changes in control to be less frequent than external changes occurring in equally uncertain environments²³⁻²⁵. ”

Last, we have extended the last paragraph of the discussion section (pp. 21-22) to explain more precisely the possible implications of our findings for two important ongoing lines of research: 1. the characterization of dysfunctions of decision-making under uncertainty in obsessive-compulsive

disorder, and 2. the understanding of biases in individual and collective decision-making in the general population:

“The increased stability of uncertain beliefs in the presence of control bears important consequences for understanding psychiatric dysfunctions of learning and decision-making under uncertainty. In particular, the description of obsessive-compulsive disorder (OCD) as a ‘doubting disease’ fails to distinguish between uncertainty about the cause of external events and uncertainty about the consequence of one’s own actions. OCD patients may show a selective impairment of outcome-based inference (when inference concerns the uncertain outcomes of their actions), but no alteration of cue-based inference (when inference concerns the uncertain source of observed stimuli). Even in the general population, the perceived degree of control over uncertain environments may explain variations in the attitude of individual decision-makers to risk over short time scales (by underestimating the true volatility of uncertain environments in the presence of control), but also variations in the accuracy of collective decisions (by increasing the perceived conflict between beliefs across individuals when they concern the expected outcomes of their actions).”

We hope that these several additions to the discussion section have clarified the conceptual significance of our findings. Again, we thank the reviewer for prompting these changes which we think have strengthened the manuscript.

2. Confounding factors.

Apart from control over information sampling, both tasks differ along another dimension: the cue-based condition entails judging a *previous* evidence sequence; the outcome-based conditions entails drawing the category for the *subsequent* evidence sequence. Can the authors rule out that this pre vs. post difference, rather than control over sampling, is the decisive factor accounting for the observed difference in perceived stability? If not, what would this mean for the broader implications? This issue needs to be discussed.

The reviewer is correct that, and despite the precise matching of the two conditions (including the instructions provided to the participants), the cue-based and outcome-based conditions may be interpreted by participants as a form of retrospective inference in the cue-based condition (judging the category of the previous sequence), and a form of prospective inference in the outcome-based condition (predicting the category of the next sequence by selecting the corresponding action).

In terms of modeling, and as described in the methods section, predicting the hidden state on the next trial corresponds in practice at trial t to choosing according to the sign of the prior log-odds belief $\mathcal{F}(L_t)$ carried over at trial $t + 1$. However, by the above expression of $\mathcal{F}(L_t)$, we can guarantee that $\mathcal{F}(L_t) > 0$ as long as $L_t > 0$ and $h < 1/2$, such that it is formally equivalent to cast choices based either on the value of L_t or $\mathcal{F}(L_t)$ as long as participants do not feature response selection variability (which they don’t, in either condition, as reported in the results section and now supported by Supplementary Fig. 1). From a modeling point of view, we can thus safely use the same Bayesian inference model to compare the two conditions.

However, in terms of cognition, the prospective nature of outcome-based inference may nevertheless play a role in the observed stabilization of hidden-state inference. To test whether the lower perceived hazard rate observed in the outcome-based condition is due to the prospective nature of inference in this condition rather than to the presence of control (as we hypothesize), we have analyzed behavioral data from a new experiment tested in $N = 30$ additional participants (5 excluded due to chance-level performance in more than one block) where we compared two variants of the cue-based condition: 1. a 'retrospective' condition in which participants were asked to report the category from which the previous stimulus sequence was drawn, and 2. a 'prospective' condition in which the same participants were asked to report the category from which the next stimulus sequence will be drawn. Importantly, unlike the outcome-based condition, the prospective condition does not confer any control to participants. But like the outcome-based condition, this new condition requires performing inference about the upcoming stimulus sequence (and its associated hidden state). Importantly, unlike the contrast between cue-based and outcome-based conditions, the contrast between retrospective and prospective conditions did not yield significant differences in reversal learning (illustrated in a new Supplementary Fig. 5).

Based on the reviewer's comment, we have added a new subsection to the results section, entitled **No difference between prospective and retrospective cue-based inference** (pp. 9-10), where we describe the results of this control behavioral study:

“The cue-based and outcome-based conditions differ in the degree of control conferred to participants over the sampling of information (instrumental control in the outcome-based condition, no control in the cue-based condition), but not only. Indeed, outcome-based inference concerns the consequences of an action – a form of 'prospective' inference, whereas cue-based inference concerns the cause of presented stimuli – a form of 'retrospective' inference. To determine whether observed differences are due to the prospective nature of inference in the outcome-based condition (rather than the control conferred to participants), we tested another group of participants ($N = 25$) in two variants of the cue-based condition: 1. a 'retrospective' condition in which participants were asked to report the category from which the previous stimulus sequence was drawn, and 2. a 'prospective' condition in which the same participants were asked to report the category from which the next stimulus sequence will be drawn. Unlike the outcome-based condition, the prospective condition does not confer any instrumental control to participants. But like the outcome-based condition, this new condition requires performing inference about the upcoming stimulus sequence.

The contrast between retrospective and prospective conditions did not yield significant differences in reversal learning (Supplementary Fig. 5a). Unlike the contrast between cue-based and outcome-based conditions, participants showed nearly identical reversal time constants in the two conditions (Supplementary Fig. 5b; retrospective: 0.676 ± 0.100 , prospective: 0.678 ± 0.131 , $t_{24} = 0.1$, $p = 0.986$). Plotting the associated response repetition curves confirmed the absence of leftward shift in the prospective condition (Supplementary Fig. 5c). Importantly, the amount of conflicting evidence required to switch did not differ between these two conditions

(Supplementary Fig. 5d; retrospective: 0.852 ± 0.144 , prospective: 1.061 ± 0.144 , $t_{24} = 1.7$, $p = 0.109$). The absence of difference between retrospective and prospective inference in the absence of instrumental control supports the notion that it is the presence of instrumental control conferred to participants in the outcome-based condition – not the prospective nature of outcome-based inference – that triggers the effects described above.”

As mentioned above, these important control analyses are now illustrated in a new Supplementary Fig. 5, whose layout matches Fig. 2 for comparison with the contrast between cue-based and outcome-based conditions:

Supplementary Fig. 5 – Reversal learning behavior during retrospective and prospective cue-based inference. Results obtained from an additional behavioral dataset ($N = 25$) in the cue-based condition where participants are instructed to report either the category of the current sequence (retrospective condition) or the next sequence (prospective condition). **a**, Response reversal curves. The thin dotted line indicates the position of the reversal. Dots indicate the observed data (means \pm s.e.m.), whereas lines indicate best-fitting saturating exponential functions. **b**, Best-fitting parameters of saturating exponential functions in the retrospective and prospective conditions. Left: the reversal time constant is not slower and the asymptotic reversal rate is not higher in the prospective condition. Black dots and error bars indicate group-level means \pm s.e.m., whereas colored dots indicate participant-level estimates. Right: correlation between psychometric parameters. As in the main experiment, the reversal time constant and asymptotic reversal rate correlate positively across tested participants. **c**, Response repetition curves. The thin dotted line indicates perfectly uncertain (null) evidence. Lines indicate best-fitting sigmoid functions. **d**, Best-fitting parameters of sigmoid functions in the retrospective and prospective conditions. Left: the PSE does not increase in the prospective condition. Right: correlation between psychometric parameters. As in the main experiment, the PSE and sensitivity to evidence correlate weakly across tested participants. *n.s.* indicates a non-significant effect.

We believe that these results strengthen further our conclusions that it is the control conferred to participants which decreases the perceived hazard rate in the outcome-based condition.

3. Rationale for MEG searchlight decoding analysis.

The searchlight decoding analysis is carried out in a way that shows only regions, in which a single variable can be decoded better than others. In other words, regions which exhibit similar decoding precision of multiple variables get under the radar.

The theoretical justification of this choice is unclear. It seems to be driven by a general desire to present a picture of neat regional specialization (e.g., p. 17, 2nd par: "To increase the spatial selectivity of obtained results..."). Cortical regions commonly contain mixtures of neurons selective for multiple different variables, not a single one. Indeed, even single neurons commonly encode multiple variables. That does not mean that these regions / neurons are irrelevant for the transformation of any particular variable. The authors should show all regions that exhibit significant decoding of any variable, irrespective of decoding of other variables.

Note that the above approach not only misses an important aspect of cortical function, it also limits the of conclusions that can be drawn from the results. Specifically, I don't think that the conclusion (p. 17, same par.) "Together, these results highlight a spatiotemporal cascade of neural patterns involved in the processing of each stimulus, ..." is valid; supporting this conclusion would require showing full maps and time courses of decoding for each variable, irrespective of the others.

Indeed, the reviewer is correct that our analyses of the coding of stimulus characteristics in MEG signals sought to identify the brain regions in which one of the stimulus characteristics can be decoded more precisely than the others. We originally did not choose to include brain maps showing all regions that show significant decoding of each characteristic, irrespective of the decoding of other characteristics. We do not mean to suggest that the identified brain regions have pure selectivity to a single stimulus characteristic, only that they are *more* selective to one characteristic than the others. Based on the reviewer's comment, we now provide such maps in a new supplementary figure (Supplementary Fig. 8a). Due to the linear mixing of sensor data used to reconstruct activity at each cortical source, we could decode each stimulus characteristic better than chance at a liberal statistical threshold (uncorrected $p < 0.05$) at all cortical sources (Supplementary Fig. 8b for five regions of interest). Given these limitations of source-reconstructed MEG signals for conducting 'searchlight' decoding analyses, we now describe more explicitly in the results section the motivation for conducting selectivity-based analyses (p. 13):

"Next, we sought to identify the cortical distribution of these shared neural codes across conditions. For this purpose, we estimated the cortical sources of observed MEG signals and performed focal multivariate pattern analyses in source space using a 'searchlight' approach (see Methods). Due to the linear mixing of sensor data used to reconstruct activity at each cortical source, we could decode each stimulus characteristic better than chance at all cortical sources (Supplementary Fig. 8). However, the coding precision of the same characteristic varied greatly across cortical sources, and in different ways for the three characteristics. Therefore, to increase the spatial selectivity of obtained results, we estimated the cortical sources for which the coding precision of one stimulus characteristic significantly exceeds the coding precision of the other two characteristics (Supplementary Fig. 9)."

We nevertheless believe that it is appropriate to describe the obtained results as a "*spatiotemporal cascade of neural patterns involved in the processing of each stimulus*" in the discussion section.

Indeed, as illustrated in a new Supplementary Fig. 9, the searchlight-based analysis of selectivity reveals a clear rostro-caudal gradient of selectivity to stimulus characteristics (Supplementary Fig. 9c), from early occipital selectivity to stimulus orientation (red-shaded area on the left) to late frontal selectivity to stimulus evidence (blue-shaded area on the right):

Supplementary Fig. 9 – Searchlight-based selectivity to stimulus characteristics across the cortical surface. a, Normalized time courses of neural coding of stimulus orientation (red), stimulus change (green) and stimulus evidence (blue) in five regions-of-interest. The neural coding of each stimulus characteristic is normalized by its global coding across time points and cortical sources to correct for large differences between stimulus characteristics. Selectivity is defined as higher normalized coding of a stimulus characteristic than the other two characteristics. All five regions-of-interest show simultaneous coding of the three stimulus characteristics, but to different extents over time. Lines and shaded error bars indicate jackknifed group-level means \pm s.e.m. **b**, Temporal clusters of selectivity to stimulus characteristics in the five regions-of-interest, corrected for multiple comparisons at the cluster level at $p < 0.001$. V1 shows early selectivity to stimulus orientation at 100 ms following stimulus onset. The TPJ shows later selectivity to stimulus change at 200 ms. The IPS shows selectivity to stimulus change at 200 ms, and selectivity to stimulus evidence after 350 ms. The LPFC and insula show selectivity to stimulus evidence after 350 ms. **c**, Rostro-caudal gradient of selectivity to stimulus characteristics. Dots indicate peaks of normalized coding of each stimulus characteristic (red: orientation, green: change, blue: evidence) for each cortical source ($N = 5,000$) along the rostro-caudal axis, from occipital cortex (left) to frontal cortex (right). Lines and shaded error bars indicate jackknifed means \pm s.e.m. of coding peaks for each stimulus characteristic along the rostro-caudal axis, obtained using robust spline smoothing. Coding peaks show a rostro-caudal gradient of selectivity to stimulus characteristics, from occipital selectivity to stimulus orientation (red-shaded area on the left) to frontal selectivity to stimulus evidence (blue-shaded area on the right). The vertical dashed line indicates the position of the central sulcus on the rostro-caudal axis. Abbreviations: V1 for primary visual cortex, TPJ for temporoparietal junction, IPS for intraparietal sulcus, and LPFC for lateral prefrontal cortex.

Finally, to measure the degree of collinearity between the neural codes of different stimulus characteristics, we have analyzed the covariations between the decoded value of one stimulus characteristic and another stimulus characteristic, resulting in the 3-by-3 array depicted in the new Supplementary Fig. 7. As can be seen, the neural code of each characteristic (on-diagonal panels) mostly evolves through ‘null’ dimensions for the other two characteristics (off-diagonal panels). This means that even though the neural codes of the different stimulus characteristics show a large degree of spatial overlap in searchlight-based analyses (Supplementary Fig. 8), the neural codes are not highly collinear. We believe that these additional analyses and results provide a more complete picture of the neural coding of stimulus characteristics. And importantly, these additional analyses confirm that these neural codes are strikingly identical (shared) across the cue-based and outcome-based conditions.

4. Neural "consistency signal".

a. Much of the discussion focusses on the MTL result. Apart from my doubt about the general approach (see point 3), it is unclear why a neural signal measuring consistency between current belief and new evidence matters for the process under study: this quantity plays no role in the decision computation described on pages 40-41.

Based on comments from the different reviewers, we realize how our description of the possible function for the consistency signal in the original manuscript may be misleading. The consistency between the current belief and the incoming evidence matters a lot for studying hidden-state inference. Indeed, as we now stress in the results section of the revised manuscript (p. 13), “*inference requires updating the prior belief in the current value of the hidden state [...] with the evidence provided by each stimulus in the new sequence*”. By studying the neural patterns involved in this interaction between incoming evidence and prior beliefs, and irrespective of its precise functional role in the decision process, we sought to “*identify the brain signatures of the stabilization of hidden-state inference in the outcome-based condition*”. In practice, as now explained in the results section of the revised manuscript (p. 15):

“We reasoned that the neural coding of the consistency of evidence with ongoing beliefs offers leverage to validate the lower perceived hazard rate in the outcome-based condition. Instead of computing belief trajectories based on the hazard rate h^* fitted to participants’ behavior (which was thus lower in the outcome-based condition), we estimated separately in each condition and for each participant the ‘neural’ hazard rate for which we could best decode consistency from MEG signals – ignoring its best-fitting value obtained from behavior (Fig. 6c).”

The analysis of the trajectories of hidden-state beliefs through the neural coding of consistency in sequences leading to response reversals (Fig. 6d) is also highly supportive of the stabilization of hidden-state inference in the outcome-based condition. Indeed, “*coding switches should occur later inside a sequence in the outcome-based condition where prior beliefs are larger and thus require more conflicting evidence to switch sign*” (p. 15), a neural prediction supported by the MEG data.

Regarding the possible functional role of the consistency signal in the decision process, we have updated the discussion section to stress that this relational code may actually reflect a dynamic prediction error signal, which can be used by ‘temporal difference’ learning algorithms to update the current belief based on the evidence provided by each new stimulus (p. 21):

“ In terms of computations, this relational code corresponds to a contextual, history-dependent variable, not to a momentary evidence variable independent of ongoing beliefs^{7,39–41}. Indeed, its reference point (i.e., the current belief) can switch sign within a single sequence of stimuli. In the outcome-based condition, this means that a non-target stimulus is coded negatively at the beginning of a sequence, but another can be coded positively later in the same sequence if a change of mind has occurred between the two stimuli – unlike classical descriptions of reward prediction errors in the midbrain^{42,43}. Such dynamic coding scheme is however consistent with recent accounts of opposite coding of factual and counterfactual outcomes in the frontal cortex⁴⁴, and of hidden-state inference by dopaminergic signals⁴⁵. Another property of this relational code, shared with other prediction error signals, is that it scales with the magnitude of the current belief: it is almost not expressed when the current belief is weak, but strongly expressed when the current belief is strong. This consistency signal can thus be described as a dynamic prediction error, tied to the current belief and not to the previous decision, which can be used by canonical ‘temporal difference’ learning algorithms^{5,6} in both conditions to update the current belief following each new piece of evidence. ”

We had originally considered the dynamic nature of this prediction error signal (the fact that it is tied to the current belief, which can change sign in the middle of a sequence – see our response to the next comment 4b) as “*impractical for downstream read-out*”, but Reviewer #3 has correctly stated that it is not necessarily the case for an inference process whose reference corresponds to its current belief rather its previous response. We hope that these changes and clarifications outline how the study of the consistency signal is relevant for understanding the difference between cue-based and outcome-based conditions.

b. The progression from Fig. 5 to 6, and the associated text, are also confusing. In light of the dynamic belief updating model used here, the obvious way of quantifying consistency is by means of the "belief trajectory analysis" from Fig. 6. This uses the instantaneous (transformed) belief state before each new cue as prior, against which consistency of the evidence is assessed. The analysis from Fig. 5 uses only the prior at the start of trial, thus ignoring possible changes of belief state during the course of the trial. There is a clear conceptual advantage for the former analysis (Fig. 6) over the latter (Fig. 5) – Fig. 5 thus seems like an unnecessary detour. It's not clear what this (suboptimal) analysis adds to the story. I suggest dropping this altogether.

We thank the reviewer for raising this point, shared with other reviewers. We have reshaped the results section by removing the main figure focusing on consistency relative to the prior belief at the start of the current trial, and replacing it by a new Supplementary Fig. 10 which clarifies why the distinction between the two definitions of consistency (relative either to the prior belief at the beginning of the trial, or to the current belief at the onset of the current stimulus) is important:

Supplementary Fig. 10 – Neural evidence for stimulus-level hidden-state inference. **a**, Estimated coding similarity for stimulus consistency across conditions. Cross-condition generalization indicates near-perfect similarity across conditions (jackknifed mean: 95%). Bars and error bars indicate group-level means \pm s.e.m. Black dots show best-fitting values from the similarity estimation procedure. **b**, Alternative update schemes for hidden-state inference indistinguishable from behavior alone. Top row: stimulus-level inference. This scheme assumes that participants update their belief in the current value of the hidden state after each stimulus, throughout each sequence of stimuli. In this same example, the change-of-mind now occurs midway through the sequence, as soon as a conflicting stimulus (here, stimulus 2) flips the sign of the log-odds belief. Under this stimulus-level update scheme, stimulus consistency is defined as the evidence provided by each stimulus in favor of the current belief accounting for previous stimuli in the same sequence. Bottom row: response-level inference. This scheme assumes that participants update their belief in the current value of the hidden state when probed for a response, following each sequence of stimuli. In this example, a change-of-mind occurs when participants combine their prior belief with the evidence provided by the current sequence. Under this response-level update scheme, stimulus consistency is defined as the evidence provided by each stimulus in favor of the previous response – which reflects the prior belief at the beginning of the current trial. Positive consistency indicates evidence consistent with the previous response (such as stimulus 1), whereas negative consistency indicates evidence conflicting with the previous response (such as stimuli 2 and n). As can be seen, the two alternative definitions of stimulus consistency differ for all stimuli presented after a mid-sequence belief switch – such as stimulus n in this example. **c**, Left: neural coding of stimulus consistency assuming stimulus-level inference (solid lines) and response-level inference (dashed lines). Stimulus consistency is coded more precisely in relation to the current belief (stimulus-level inference) than to the prior belief (response-level inference). The shaded area indicates the significant difference in coding precision between the two update schemes. Lines and shaded error bars indicate group-level means \pm s.e.m. of coding precision, baseline-corrected across conditions using the last 100 ms preceding stimulus onset (dashed area) to account for possible non-zero correlations across successive stimuli. Right: neural coding of stimulus consistency (solid lines) and the magnitude of the current belief (dashed lines). Stimulus consistency is coded in a stronger and more sustained fashion than the magnitude of the current belief. The shaded area indicates the significant difference in coding precision between the two quantities.

As can be seen on Supplementary Fig. 10b, consistency defined relative to the current belief assumes that participants update their hidden-state belief after each new stimulus, throughout each

sequence – a ‘stimulus-level’ inference scheme. By contrast, consistency defined relative to the prior belief at the beginning of the trial assumes that participants update their hidden-state belief only when probed for a response, at the end of each sequence – a ‘response-level’ inference scheme. We now explain in the results section (p. 15) how the MEG data (but not behavior alone) afford to arbitrate between these two inference schemes:

“Distinguishing between stimulus-level and response-level inference schemes is not possible from behavior alone, since their applications result in the same behavior. However, the two inference schemes are associated with different belief trajectories over the course of each sequence. In particular, stimulus-level inference predicts that changes of mind can occur midway through a sequence, as soon as a conflicting stimulus flips the sign of the log-odds belief. As a result, the two inference schemes are associated with different consistency variables, whose coding precision in MEG signals can be compared. This comparison revealed a stronger coding of consistency predicted by stimulus-level inference from 150 ms following stimulus onset (Supplementary Fig. 10c; peak $F_{1,23} = 28.2$, cluster-level $p < 0.001$). This result indicates that participants update their beliefs regarding the current value of the hidden state after each stimulus – at a sub-second time scale – not only when probed for a response.”

We strongly believe that characterizing the inference scheme used by participants is important for understanding the difference between cue-based and outcome-based inference. We thank the reviewer for prompting this reshaping of the results section, which we think clarified how the MEG analyses and results complement the main behavioral finding. As a result, the results section does not make the “unnecessary detour” pointed by the reviewer, and uses MEG analyses to support the stabilization of hidden-state inference observed in the outcome-based condition.

Minor:

- The first paragraph of Introduction is confusing and misleading. It refers to a review by Shadlen & Shohamy that compares the information sources for perceptual and value-based decisions: sensory environment vs. memory. The authors move on to claim that the key differences between perceptual and value-based decisions lies in the degree of control over information sampling. This presentation conflates two separate dimensions along which perceptual and value-based decisions differ (information source and control over sampling). It also prompts the reader to expect a direct comparison between the impact of these two dimensions, which this study clearly does not deliver. The paragraph should be re-written.

We have modified the first paragraph of the introduction section accordingly. We now focus much more explicitly on the difference in the degree of control over information sampling (p. 3):

“Making accurate decisions in an uncertain environment requires inferring its properties from imperfect information^{1,2}. When categorizing an ambiguous stimulus, this ‘hidden-state inference’ process consists in identifying the generative cause of sensory cues. By contrast, when foraging rewards, inference concerns the expected outcomes of possible courses of action. A constitutive, yet rarely considered, difference between these two forms of inference lies in the degree of control over information sampling conferred to the decision-maker. Indeed, cue-based inference relies

on the presentation of information to an observer interpreting a relevant property of its environment (here, the category of the presented stimulus), whereas outcome-based inference relies on the active sampling of information by an agent interacting with its environment to achieve a particular goal (here, maximizing rewards). ”

We agree that it is important to focus the introduction section on the difference between cue-based and outcome-based inference that we can study using our experimental protocol.

- The data and model predictions look compelling; yet the authors should present more evidence that model fits the data well.

We have followed the reviewer’s advice and now present several additional results which support the idea that the model fits the behavioral data well. In Supplementary Fig. 1b, we show the results of the factorized Bayesian model selection procedure for characterizing the sources of behavioral variability in the cue-based and outcome-based conditions. This analysis (validated by a model recovery procedure) shows that participants feature inference noise but no response selection noise in the two conditions. Supplementary Fig. 1cd shows how the predictions of noisy inference match the decreasing sensitivity to evidence with sequence length (see Drugowitsch, Wyart et al., 2016, *Neuron*), whereas the predictions of noisy response selection do not match the same data. Supplementary Fig. 1e shows that the two conditions do not differ in terms of the subjective weighting of stimuli, or the temporal integration kernels within each sequence of stimuli. In other words, participants do not show more leaky inference within each sequence in the cue-based condition – something which could masquerade as a lower perceived hazard rate. These additional analyses provide additional evidence that the Bayesian inference model fits the human data well.

- How could the repetition effects in Fig. 2 be produced by any parameter in the model used, other than subjective hazard rate? The claim that this result "offers a computational account for the increased PSE in the outcome-based condition" seems overstated.

To provide supporting evidence that the increased PSE in the outcome-based condition arises from a lower perceived hazard rate, we have fitted the human data with a ‘descriptive’ model of hidden-state inference – illustrated in a new Supplementary Fig. 2. In contrast to the ‘normative’ model used in the main text, the descriptive model does not fit the perceived hazard rate (the hazard rate being a generative parameter of the task), but an empirical prior function $\mathcal{F}^*(L_{t-1})$ with two separate parameters: 1. a scaling term α which corresponds to the fraction of the posterior belief L_{t-1} at the end of the previous trial used as prior belief at the beginning of the current trial, and 2. a constant term δ which corresponds to a form of repetition bias (Supplementary Fig. 3a). Due to the saturating nature of the normative prior function, the constant term in the empirical prior function is expected to be significantly positive – which it is (Supplementary Fig. 3b). However, it only grows by 20% in the outcome-based condition. By contrast, as expected by an increased transfer of the hidden-state

belief between consecutive trials, the scaling term shows a nearly two-fold increase (97%) in the outcome-based condition.

We also provide in Supplementary Fig. 3 additional ‘model-free’ evidence that a larger fraction of the posterior belief is retained as prior belief at the onset of the next trial in the outcome-based condition. For this purpose, we split the behavioral data between trials where the evidence provided by the previous sequence in favor of the previous response is either weak (smaller than its median value) or strong (larger than its median value). By doing so, we split the behavioral data between trials where the prior belief is either weak or strong. A repetition bias predicts that the PSE on the current trial should not depend on this split. By contrast, a perceived hazard rate $h < 0.5$ predicts that the PSE on the current trial should be larger when the prior belief is strong. The behavioral data shows a strong increase in the PSE as a function of this split (Supplementary Fig. 3d), which indicates that the data in the two conditions are better explained by a perceived hazard rate $h < 0.5$ than by a repetition bias (Supplementary Fig. 3e). Furthermore, this increase in the PSE is significantly larger for the outcome-based condition, which is consistent with our claim that the perceived hazard rate is lower in the outcome-based condition.

We believe that these additional analyses and results, now mentioned in the results section (p. 9) and illustrated in Supplementary Fig. 3, provide more compelling evidence that a lower perceived hazard rate provides a “*computational account of the increased PSE in the outcome-based condition*”:

“Additional analyses confirmed that a larger fraction of the posterior belief is carried over to the next trial in the outcome-based condition, consistent with a lower perceived hazard rate but at odds with an increased response repetition bias (Supplementary Fig. 3).”

| - Are all results except Fig. 3e collapsing across the two objective hazard rates? This needs to be explicitly stated.

Indeed, all results presented in the main text (except Fig. 3e) collapse across ‘more stable’ and ‘more volatile’ blocks. We now state this explicitly in the methods section. We have also added a new Supplementary Fig. 4, and additional text in the results section (p. 9), to explain why we have chosen to collapse the two conditions in the main analyses and to provide the results of separate model fits to each type of blocks:

“ This difference in perceived hazard rate between conditions makes a testable prediction: participants should be more accurate in the outcome-based condition in more stable environments where reversals are rare, and more accurate in the cue-based condition in more volatile environments where reversals are frequent (Fig. 3e). Unbeknownst to participants, we varied the true hazard rate across blocks between 0.083 (more stable) and 0.167 (more volatile). As predicted, participants were more accurate in the outcome-based condition in more stable blocks ($t_{23} = 3.4$, $p = 0.002$), and more accurate in the cue-based condition in more volatile blocks ($t_{23} = -2.5$, $p = 0.020$, interaction: $F_{1,23} = 20.2$, $p < 0.001$). This interaction is driven by the fact that participants

did not adapt their perceived hazard rate to these fine, uncued changes in true hazard rate (Supplementary Fig. 4a, b), with perceived hazard rates closer to the true hazard rate for the outcome-based condition in more stable blocks, and for the cue-based condition in more volatile blocks. Importantly, the decrease in perceived hazard rate in the outcome-based condition was highly similar for participants with more stable inference (i.e., low perceived hazard rates) and participants with more volatile inference (i.e., high perceived hazard rates) across conditions (Supplementary Fig. 4c, d). This pattern of findings supports the conclusion that participants perceive the outcome-based condition as more stable than the cue-based condition, despite identical true hazard rates. ”

- The objective hazard rates should be plotted along with the subjective ones in Fig. 3b, for reference. I realized that there are two h values intermixed; perhaps then present the pooled h – after all, the behavioral modeling also seems to pool across these two contexts.

We have followed the reviewer’s suggestion. Instead of pooling the true hazard rate across the two types of blocks in Fig. 3b, we have chosen to present the true hazard rates together with the best-fitting values of the perceived hazard rates separately for the ‘more stable’ and ‘more volatile’ blocks in Supplementary Fig. 4 where we perform separate model fits for the two types of blocks.

Reviewer #2

In this paper the authors examine how people detect changes in the state of the environment during a reversal learning task. In this task sensory stimuli are drawn from one out of two categories with the generative category changing infrequently and at random points. Unlike previous studies, here the authors compare behaviour in two normatively equivalent conditions. In the first condition (“cue-based”), participants first observe and then categorise the sensory stimulus. In the second condition (“outcome-based”) participants make a prediction about the categorical identity of the stimulus in the forthcoming trial. The authors report that in the outcome-based condition people are more prone to miss switches in the generative state of the environment, which translates into a larger choice repetition rate. These results are attributed to a stronger reliance on prior information, which in turn falls out from an overestimation of the stability of the environment. Further analyses using magnetoencephalography (MEG) confirm and further corroborate the behavioural results.

Overall, this is a very well-designed experiment addressing the important question of belief updating in relation to the way people interact with information. The reported behavioural effect appears to be robust, however an explanation of this effect is currently missing. Why do people rely on prior information more strongly in the outcome-based condition and does this necessarily imply a larger sense of stability? Below I outline the points of criticism in detail.

We thank the reviewer for this positive assessment of the novelty and potential importance of the question and finding. In addition to addressing the reviewer’s specific comments, we have also tried to clarify in the introduction and discussion sections of the revised manuscript why this result should matter to existing theories of learning and decision-making, both in the general population and in specific psychiatric disorders.

First, we have revised the discussion section to explain earlier and more prominently the implications of our findings. The issue of whether and how interacting with one’s environment affects how we learn from it has been a hugely influential topic in the learning sciences for decades, something which we now stress explicitly in the discussion section (p. 19):

“A first immediate conclusion is that the status of the decision-maker – an agent interacting with its environment in the outcome-based condition, or an observer contemplating the same environment in the cue-based condition – shapes human learning and decision-making under uncertainty. In the learning sciences, the distinction between ‘active’ and ‘passive’ information gathering has been stressed for decades¹¹. Controlling the flow of incoming information during learning is seen as an important degree of freedom for acquiring non-redundant information and accelerating the learning process. By contrast, and surprisingly, existing theories of cognitive inference either do not consider the degree of control over the sampling of evidence as a contextual variable^{2,4,7,15}, or formulate distinct computational models of ‘active’ and ‘passive’ inference¹⁸. ”

We now also explain earlier in the discussion section (pp. 19-20) why and how the presence of control may trigger the observed stabilization of hidden-state inference. By discussing the possible driving forces of the observed effect, this paragraph offers possible implications for our findings:

“ Interacting with uncertain environments confers several cognitive advantages, which could explain the lower perceived hazard rate and the resulting stabilization of hidden-state inference in the out-come-based condition. First, the active sampling of evidence enables testing current hypotheses in a self-directed, online fashion – an ability known to improve human learning in novel environments^{11,12}. A functional interpretation is that the lower perceived hazard rate in the out-come-based condition aims at testing the current hypothesis that the previous action samples from the target category. Such hypothesis testing is by definition impossible in the cue-based condition because participants had no control over the sampling of evidence. Note that the inference process used in the cue-based condition is not equivalent to ‘observational’ learning – i.e., learning from or about another agent interacting with uncertain evidence^{19,20}. Indeed, participants were actively committing to interpretations regarding the source of evidence in both conditions, the sole difference being the presence or absence of instrumental control over the sampling of evidence. Even without considering hypothesis testing, knowledge about action-outcome contingencies affords an agent to predict incoming sensory signals, and to stabilize these signals by stabilizing its behavior^{18,21,22}. The lower perceived hazard rate in the outcome-based condition may thus aim at stabilizing the source of the incoming evidence – something which is not possible in the absence of control. Last, the difference in perceived hazard rate between the two conditions may betray a temporal hierarchy between two types of changes which can arise in uncertain environments: 1. external (uncontrollable) changes in the state of the environment (as in the cue-based condition), and 2. changes in the instrumental control of the same environment (as in the outcome-based condition). In this view, participants behave as if they assume changes in control to be less frequent than external changes occurring in equally uncertain environments^{23–25}. ”

Last, we have extended the last paragraph of the discussion section (pp. 21-22) to explain more precisely the possible implications of our findings for two important ongoing lines of research: 1. the characterization of dysfunctions of decision-making under uncertainty in obsessive-compulsive disorder, and 2. the understanding of biases in individual and collective decision-making in the general population:

“ The increased stability of uncertain beliefs in the presence of control bears important consequences for understanding psychiatric dysfunctions of learning and decision-making under uncertainty. In particular, the description of obsessive-compulsive disorder (OCD) as a ‘doubting disease’ fails to distinguish between uncertainty about the cause of external events and uncertainty about the consequence of one’s own actions. OCD patients may show a selective impairment of outcome-based inference (when inference concerns the uncertain outcomes of their actions), but no alteration of cue-based inference (when inference concerns the uncertain source of observed stimuli). Even in the general population, the perceived degree of control over uncertain environments may explain variations in the attitude of individual decision-makers to risk over short time scales (by underestimating the true volatility of uncertain environments in the presence of control), but also variations in the accuracy of collective decisions (by increasing the perceived conflict between beliefs across individuals when they concern the expected outcomes of their actions). ”

We hope that these several additions to the discussion section have clarified the conceptual significance of our findings. Below we have also addressed each of the reviewer's specific comments, which have clarified our findings and their significance.

1) There is an overall larger choice repetition rate in the outcome based condition. This effect can be a global effect, equally present in trials of all type, or driven by repetition behaviour in specific trials. The current analysis presented in Figure 2b is a first step towards fully characterising the behavioural differences between the two conditions. However, I recommend that the authors go one step further and perform this analysis by splitting trials in "repetition correct" and "repetition incorrect". This will shed light on whether participants tend to have different repetition rates when a switch has occurred.

We have conducted the exact analysis suggested by the reviewer, illustrated in Supplementary Fig. 2a. We found no difference in the choice repetition psychometric curves between trials where the hidden state does not change ('repetition correct') and trials where the hidden state reverses ('repetition incorrect'). This absence of difference suggests that the larger PSE observed in the outcome-based condition corresponds to a 'global' effect – as predicted by the lower perceived hazard rate in our Bayesian inference model. We have also checked that the larger PSE in the outcome-based condition is not only present for specific sequence lengths in Supplementary Fig. 2b. We found that it is not the case: the difference in PSE between cue-based and outcome-based conditions is present (and equally strong) across all tested sequence lengths – from 2 to 8 stimuli. These additional analyses suggest that response reversals are driven predominantly by the interaction between prior beliefs and the incoming evidence provided by the current stimulus sequence, as assumed by our Bayesian inference model depicted on Fig. 3.

2) Participants in the outcome-based condition have an overall smaller perceived hazard rate. Since this is the central finding in the paper, this analysis needs to be further unpacked. First, how does the signed and unsigned difference between the perceived hazard rate and the actual hazard rate differ as a function of the actual hazard rate in the two conditions? Even if people underestimate the hazard rate in the outcome-based condition their estimate could still be closer to the actual hazard rate. Second, how do individual differences look like with regards to the over /under-estimation of hazard rate? For instance, do participants consistently overshoot/ undershoot the hazard rate in both conditions? This more detailed analysis will allow understanding the changes in behaviour between outcome-based and cue-based conditions.

We have performed several additional analyses, illustrated in Supplementary Fig. 4, which allow to further characterize the behavioral differences between cue-based and outcome-based conditions. Supplementary Fig. 4a depicts the perceived hazard rate estimated separately in blocks with low and high true hazard rate. As can be seen, participants did not adjust their perceived hazard rate to these subtle, uncued changes in true hazard rate, in either condition. At the group level, participants tend to overestimate the true hazard rate in the cue-based condition, whereas they tend to underestimate it in the outcome-based condition in the blocks with high true hazard rate. We ran additional

model simulations to show that the perceived hazard rate should have adjusted to the true hazard rate to maximize performance – even considering the same inference noise as participants:

Supplementary Fig. 4 – Robustness of psychometric effects to inter-individual variability. **a**, Relative insensitivity of tested participants to subtle changes in true hazard rate across blocks. Perceived hazard rate estimates in the cue-based (left) and outcome-based (middle) conditions as a function of the true/generative hazard rate (low: more stable blocks, high: more volatile blocks), marked as black lines. Black dots and error bars indicate group-level means \pm s.e.m., whereas colored dots indicate participant-level estimates. The perceived hazard rate does not adapt to changes in true hazard, despite the fact that the optimal (accuracy-maximizing) hazard rate (right) tracks changes in true hazard rate. **b**, Left: correlation between perceived hazard rate estimates in more stable (x-axis) and more volatile (y-axis) blocks. Right: correlation between inference noise estimates in more stable and more volatile blocks. Dots and error bars indicate posterior means \pm s.d. obtained by model fitting. The thin dotted line shows the identity line. Both parameters correlate strongly between blocks across tested participants. **c**, Psychometric effects for participants with more volatile inference ($N = 12$). Left: perceived hazard rate estimates. Middle: response repetition curves, showing a clear leftward shift in the outcome-based condition. Lines indicate group-level means \pm s.e.m. Right: best-fitting psychometric parameters. The PSE (left) is increased in the outcome-based condition, whereas the sensitivity to evidence (right) is equal across conditions. **d**, Psychometric effects for participants with more stable inference ($N = 12$). Left: perceived hazard rate estimates. Middle: response repetition curves, showing also a clear leftward shift in the outcome-based condition. Right: best-fitting psychometric parameters. Like the other group, the PSE (left) is increased in the outcome-based condition, whereas the sensitivity to evidence (right) is equal across conditions. Two stars indicate a significant effect at $p < 0.01$, three stars at $p < 0.001$, *n.s.* a non-significant effect.

This pattern of findings suggests that the perceived hazard rates used by participants are much more endogenously driven than exogenously driven: they show a large inter-individual variability and a strong within-participant correlation between more stable and more volatile blocks (Supplementary Fig. 4b), but no difference between more stable and more volatile blocks.

We also followed the reviewer's suggestion and split participants as a function of their mean perceived hazard rate pooled across conditions and types of blocks: 1. participants with more volatile inference (higher h parameter than its median value across participants, $N = 12$; Supplementary Fig. 4c) and 2. participants with more stable inference (lower h parameter than its median value across participants, $N = 12$; Supplementary Fig. 4d). Importantly, we found that both subgroups of participants show the same selective difference between cue-based and outcome-based conditions: a lower perceived hazard rate h in the outcome-based condition (left panel), associated with an increased PSE but no change in the sensitivity to evidence (right panels). We hope that these detailed analyses clarify the behavioral differences between conditions. Together, they support the idea of a lower perceived hazard rate in the outcome-based condition.

3) The finding that participants rely more heavily on the prior belief in the outcome-based conditions is consistent with a lower perceived hazard rate in the normative framework that the authors employ. However, stronger reliance on the prior could just be an idiosyncratic factor independent of perceived stability. In Bayesian terms that would correspond to having a more precise prior distribution in the outcome-based condition. Can this study dissociate the two possibilities? It will also be instructive to consider more naïve Bayesian models in which the hazard rate is not a free parameter.

We understand the reviewer's comment. The Bayesian inference model assumes, as indicated in the instructions provided to the participants (and provided in the methods section), that the hidden state s_t is binary – i.e., can take only two values s_1 and s_2 . Therefore, the 'binomial' belief at trial t can be summarized by a single scalar variable which corresponds to $\log(p(s_t = s_1)/p(s_t = s_2))$. As indicated in the results section (p. 7), "*the sign of the log-odds belief indicates whether s_1 or s_2 is more likely, whereas the magnitude of the log-odds belief indicates the strength of the belief in favor of the more likely hidden state*". In other words, the magnitude of the log-odds belief $\text{abs}(L_t)$ corresponds to the precision of the belief. As long as the Bayesian inference model correctly assumes that the hidden state is binary, a lower perceived hazard rate – a 'normative' description of the cognitive effect – corresponds at the 'descriptive' (process) level to a more precise prior belief, as proposed by the reviewer. As a result, we cannot distinguish in practice between these two accounts because they correspond to the same effect described either in normative or descriptive terms. The two accounts differ only if one assumes a metacognitive process which tracks the second-order certainty about the current value of the log-odds belief (which itself reflects the certainty about the current value of the hidden state). This second-order process was not required to fit the behavioral data, and was not required to fit confidence ratings obtained after each response in another (unpublished) dataset using the same experimental protocol.

Nevertheless, we have slightly modified the results section (pp. 7-9) to clarify how a decrease in the perceived hazard rate at the ‘computational’ level corresponds to more precise (i.e., larger) prior beliefs at the beginning of each trial at the ‘algorithmic’ level:

“By contrast, comparing perceived hazard rates between conditions revealed a significant decrease in the outcome-based condition (Fig. 3c; Cb: 0.191 ± 0.022 , Ob: 0.115 ± 0.015 , $t_{23} = -7.7$, $p < 0.001$). This difference offers a computational account for the increased PSE in the outcome-based condition. Indeed, the decrease in perceived hazard rate in the outcome-based condition boosts prior beliefs by 42% (Fig. 3d), thereby requiring more conflicting evidence to reverse a previous response. Additional analyses confirmed that a larger fraction of the posterior belief is carried over to the next trial in the outcome-based condition, consistent with a lower perceived hazard rate but at odds with an increased response repetition bias (Supplementary Fig. 3).”

Following the reviewer’s suggestion, we have also considered in Supplementary Fig. 3 naïve Bayesian models using a ‘descriptive’ update of beliefs following each trial (in terms of constant and linear scaling terms) rather than the ‘normative’ update (in terms of a perceived hazard rate). The descriptive (naïve) model uses two parameters to describe the update of beliefs between two consecutive trials: 1. a constant term that biases the prior belief at trial t in direction of the posterior belief at trial $t - 1$ – irrespective of its magnitude, and 2. a scaling term that scales the prior belief at trial t as a fraction of the posterior belief at trial $t - 1$ (Supplementary Fig. 3a).

We found that the scaling term shows a two-fold (97%) increase in the outcome-based condition, whereas the constant term shows only a 20% increase (Supplementary Fig. 3b). This additional analysis suggests that participants indeed retain a larger fraction of their previous belief as prior at the onset of the new trial in the outcome-based condition. This observation is highly consistent with a lower perceived hazard rate in the outcome-based condition. We further validated this result in Supplementary Fig. 3c by splitting the behavioral data between trials where the evidence provided by the previous sequence in favor of the previous response is either weak (smaller than its median value) or strong (larger than its median value). By doing so, we split the behavioral data between trials where the prior belief is either weak or strong. A repetition bias (i.e., an idiosyncratic factor independent of perceived stability) predicts that the PSE on the current trial should not depend on this split. By contrast, a perceived hazard rate $h < 0.5$ predicts that the PSE on the current trial should be larger when the prior belief is strong. The behavioral data shows a strong increase in the PSE as a function of this split (Supplementary Fig. 3d), which indicates that the data in the two conditions are better explained by a perceived hazard rate $h < 0.5$ than by a repetition bias (Supplementary Fig. 3e). Furthermore, this increase in the PSE between trials with weak vs. strong prior is significantly larger for the outcome-based condition, which is consistent with our claim that the perceived hazard rate is lower in the outcome-based condition.

We believe that these additional analyses and results, now mentioned in the results section (p. 9) and illustrated in Supplementary Fig. 3, provide more compelling evidence in favor of a lower perceived hazard rate in the outcome-based condition:

“ Additional analyses confirmed that a larger fraction of the posterior belief is carried over to the next trial in the outcome-based condition, consistent with a lower perceived hazard rate but at odds with an increased response repetition bias (Supplementary Fig. 3). ”

4) In their MEG analysis the authors focus on 3 relevant quantities pertaining to the perceived stimulus (Figure 4). However, it is not clear why they do not consider the neural correlates of the prior belief in a similar fashion. This seems to be an obvious quantity to consider, unless there are confounding factors that prevent the authors from decoding the prior belief (currently not discussed).

In our analyses of the MEG data, we have chosen to focus on quantities whose computations are tied to the presentation of a new stimulus (the sole source of evidence for the hidden-state inference process). The computation of the prior belief being not locked to the onset of a stimulus, we have not considered it in our analyses. Nevertheless, we could extract the neural correlates of the magnitude of the current belief, $\text{abs}(L_t)$, including the evidence provided by the current stimulus, locked to the onset of the current stimulus. We believe this quantity to be less informative than the consistency of the ongoing belief with the current stimulus – whose sign distinguishes between stimuli consistent with the ongoing belief and stimuli conflicting with the ongoing belief. As indicated in the results section, the neural coding of consistency provided leverage to identify belief reversals in a middle of a sequence (p. 15):

“ We reasoned that the neural coding of consistency, defined in relation to the prior belief at the beginning of the current sequence, should switch sign when the belief itself switches midway through the sequence. These mid-sequence belief switches are bound to occur in trials leading to response reversals (where the belief at the end of the trial has opposite sign to the belief at the beginning of the trial), but not in sequences leading to response repetitions (where the belief at the end of the trial has the same sign as the belief at the beginning of the trial). Furthermore, these coding switches should occur later inside a sequence in the outcome-based condition where prior beliefs are larger and thus require more conflicting evidence to switch sign. ”

We used these coding switches to visualize the more stable beliefs – corresponding to a lower perceived hazard rate – in the outcome-based condition. Nevertheless, following the reviewer’s advice, we present the time course of its neural coding in MEG signals in Supplementary Fig. 10c (dotted lines on the right panel). As can be seen, the magnitude of the current belief can be decoded from MEG signals, but in a weaker and less sustained fashion than the consistency of the current stimulus with the belief at stimulus onset.

We also tried to decode the value of the current belief, L_t , signed in direction of the associated response (positive for a hidden state s_t associated with a left-handed response, or negative for a hidden state s_t associated with a right-handed response), but this variable could not be decoded from stimulus-locked MEG signals. Extracting the band-limited power of MEG signals in the alpha band (8-16 Hz), whose transient suppression reflects a well-characterized signature of response preparation in MEG and EEG signals (see, e.g., Donner et al., 2009, *Curr. Biol.*; O’Connell et al.,

2012, *Nat. Neurosci.*; Wyart et al., 2012, *Neuron*), offered a significant decoding of the sign of the belief, $\text{sign}(L_t)$, but only in the last second preceding response execution:

Importantly, this neural coding peaks at the time of response execution, and thus likely reflects a motor signal rather than an inference signal – upstream in the decision process. It also does not differ between the cue-based and outcome-based conditions, thereby providing additional support that the increased PSE in the outcome-based condition does not reflect a repetition bias, or an idiosyncratic factor independent of perceived stability. We have chosen not to include this additional analysis in the revised manuscript, but hope that the analyses reported in Supplementary Fig. 10 and the changes to the results section clarify our focus on the neural coding of consistency.

5) The analysis in Figure 6 appears to just be a coarser version of the analysis presented in Figure 7. I would recommend merging the former with the latter or dropping the former altogether.

We thank the reviewer for this comment. We realize how our dual definition of the consistency signal in the original manuscript may be misleading. We have reshaped the results section by removing the main figure focusing on consistency relative to the prior belief at the start of the current trial (Fig. 6 in the original manuscript), and replacing it by Supplementary Fig. 10. This new figure, accompanied by new text in the results section (p. 15), clarifies why the distinction between the two definitions of consistency (relative either to the prior belief at the beginning of the trial, or to the current belief at the onset of the current stimulus) is important.

As can be seen on Supplementary Fig. 10b, consistency defined relative to the current belief assumes that participants update their hidden-state belief after each new stimulus, throughout each sequence – a ‘stimulus-level’ inference scheme. By contrast, consistency defined relative to the prior belief at the beginning of the trial assumes that participants update their hidden-state belief only when probed for a response, at the end of each sequence – a ‘response-level’ inference scheme. We now explain in the results section (p. 15) how the MEG data (but not behavior alone) afford to arbitrate between these two inference schemes:

“Distinguishing between stimulus-level and response-level inference schemes is not possible from behavior alone, since their applications result in the same behavior. However, the two inference

schemes are associated with different belief trajectories over the course of each sequence. In particular, stimulus-level inference predicts that changes of mind can occur midway through a sequence, as soon as a conflicting stimulus flips the sign of the log-odds belief. As a result, the two inference schemes are associated with different consistency variables, whose coding precision in MEG signals can be compared. This comparison revealed a stronger coding of consistency predicted by stimulus-level inference from 150 ms following stimulus onset (Supplementary Fig. 10c; peak $F_{1,23} = 28.2$, cluster-level $p < 0.001$). This result indicates that participants update their beliefs regarding the current value of the hidden state after each stimulus – at a sub-second time scale – not only when probed for a response. ”

We strongly believe that characterizing the inference scheme used by participants is important for understanding the difference between cue-based and outcome-based inference. We thank the reviewer for prompting this reshaping of the results section, which we think clarified how the MEG analyses and results complement the main behavioral finding. As a result, the results section makes it clear how we use MEG analyses to support the stabilization of hidden-state inference observed in the outcome-based condition.

Reviewer #3

I enjoyed reading this interesting study of probabilistic belief updating with and without control over sampling of information, in outcome-based and cue-based task conditions respectively. The authors found

* The outcome-based condition had a longer reversal time constant & higher reversal rate asymptote - i.e. more evidence was required to switch response (i.e. a switch 'threshold' was different but response sensitivity the same & perseveration only nonsignificantly higher in outcome condition) as the hazard rate was lower (i.e. the posterior from the last trial contributed more to the prior on next trial).

* Inference 'noise' was the same across conditions; Bayesian model selection implied no response noise was added.

* Mean accuracy & RT was no different across cue/outcome conditions, but accuracy higher in outcome condition with few reversals, and lower when more reversals, as one would expect given its low hazard rate

* Orientation & change in tilt & evidence strength were decoded from the MEG data with the same precision in and across both conditions but...

* 'Consistency' of evidence with the previous response (i.e. matching/non-matching the prior) was more precisely decoded in the outcome condition - in R MTL (~200 ms) & PPC (~400 ms)...

...probably because of the lower hazard rate (consistency with the current belief was ~equally precise across conditions)

* Coding of 'relational' evidence (to current beliefs) appeared before 'absolute' evidence & its precision was proportional to magnitude of beliefs, and greater than precision of absolute coding in MTL

The authors have conceived a great question, with a good experiment design, careful and rigorous methodology, and interesting results. The paper itself is very clearly written. Its conclusions will be relevant to numerous fields, and I agree with the authors that they may be especially relevant to psychiatric disorders and OCD in particular. Overall I support its publication, and I have only relatively minor comments:

We thank the reviewer for this very encouraging assessment of our work. We hope to have addressed each of his/her comments, and we thank him/her for insightful reflections on the potential function of the consistency signal in the hidden-state inference process. We have updated the discussion section accordingly to clarify this question, and we have also revised the introduction section to focus it more on the specific question that our experimental protocol addresses – namely, how the degree of control over information sampling influences cognitive inference. We hope that the reviewer will find the revised manuscript improved in these different aspects.

Parameter recovery looks good, given the correlations across conditions in Fig 3B and 3C. I wondered the same about model recovery: from simulated data with and without a more-or-less noisy response model, does model selection correctly detect the presence or absence of the response model?

We have directly addressed this question by performing the model recovery analysis suggested by the reviewer, now illustrated in a new Supplementary Fig. 1. The confusion matrix is fully diagonal,

which indicates that the Bayesian model selection procedure is capable of correctly detecting response selection noise and distinguishing it from inference noise. Additionally, and following guidelines developed by one of us (Palminteri, Wyart and Koechlin, 2017, *Trends Cogn. Sci.*), Supplementary Fig. 1cd shows how simulations of noisy inference match the decreasing sensitivity to evidence with sequence length (see also Drugowitsch, Wyart et al., 2016, *Neuron*), whereas simulations of noisy response selection do not match the same data. These additional analyses provide additional evidence that the human data is compatible with noisy (imprecise) inference, not with stochastic response selection (e.g., softmax) – in both conditions.

P33 - I was a bit puzzled by some of the discussion of the 'relational code'. I might have misunderstood, but to me this interesting analysis is revealing the neural correlates of model-updating in the brain, i.e. the processes that occur when priors and likelihoods interact. But actually describing these correlates as a 'relational code' and saying it is "impractical for downstream read-out" is a bit misleading to me - if the system knows its prior beliefs then is it easy to 'read-out' adjustments to these beliefs.

We thank the reviewer for this comment. It is indeed entirely justified to think that the consistency signal reflects a 'relational' code expressed in the frame of reference of the current belief. In this view, the consistency signal is not at all 'impractical for downstream read-out' as we stated in the original manuscript. We have thus modified this paragraph of the discussion section to stress that this relational code may actually reflect a dynamic prediction error signal, which can be used downstream by 'temporal difference' learning algorithms to update the current belief based on the evidence provided by each new stimulus (p. 21):

“ In terms of computations, this relational code corresponds to a contextual, history-dependent variable, not to a momentary evidence variable independent of ongoing beliefs^{7,39–41}. Indeed, its reference point (i.e., the current belief) can switch sign within a single sequence of stimuli. In the outcome-based condition, this means that a non-target stimulus is coded negatively at the beginning of a sequence, but another can be coded positively later in the same sequence if a change of mind has occurred between the two stimuli – unlike classical descriptions of reward prediction errors in the midbrain^{42,43}. Such dynamic coding scheme is however consistent with recent accounts of opposite coding of factual and counterfactual outcomes in the frontal cortex⁴⁴, and of hidden-state inference by dopaminergic signals⁴⁵. Another property of this relational code, shared with other prediction error signals, is that it scales with the magnitude of the current belief: it is almost not expressed when the current belief is weak, but strongly expressed when the current belief is strong. This consistency signal can thus be described as a dynamic prediction error, tied to the current belief and not to the previous decision, which can be used by canonical 'temporal difference' learning algorithms^{5,6} in both conditions to update the current belief following each new piece of evidence. ”

We had originally considered the dynamic nature of this prediction error signal (the fact that it is tied to the current belief, which can change sign in the middle of a sequence) as “*impractical for downstream read-out*”, but the reviewer is correct that it is not problematic for an inference process whose

reference corresponds to its current belief rather than the previous response. We hope that these changes and elaborations clarify the potential functional role of the consistency signal in the hidden-state inference process. Again, we thank the reviewer for prompting this change.

I was also puzzled by: "Another property of this relational code is that it scales with the magnitude of the current belief: it is almost not expressed when the current belief is weak, but strongly expressed when the current belief is strong – an effect which violates a basic tenet of Bayesian reasoning: the independence between the likelihood and prior terms". Unless I misunderstand, the 'relational code' reflects the encoding of new evidence given the subject's prior beliefs. In Bayesian belief updating, this new evidence will be a precision-weighted prediction error, and the precision weight will be the ratio of likelihood precision/(likelihood + prior precisions). So one *would* expect the relational code to "scale with the magnitude of [i.e. confidence in] the current belief" and this certainly doesn't violate Bayes (I'm not sure about circular inference).

As discussed in our response to the previous point, the reviewer is correct that the consistency signal can be used by a dynamic inference process whose reference is tied to its current belief rather than the previous response – which is precisely the 'stimulus-level' inference scheme supported by the MEG data (see Supplementary Fig. 10). Now, as stated above and discussed in the discussion section of the revised manuscript (p. 21), the consistency signal can indeed be described "*as a dynamic prediction error, tied to the current belief and not to the previous decision, which can be used by canonical 'temporal difference' learning algorithms in both conditions to update the current belief following each new piece of evidence*". In other words, we agree with the reviewer's comment and have removed the misleading statements from the discussion section. As for the previous comment, we thank the reviewer for prompting this change to the manuscript.

Another question I have about this conclusion is whether this effect is some artefact of the training process? If there are not many reversals then presumably the classifier is trained on lots of trials when the subject has stronger beliefs, and not so many when the subjects are unsure. So the relational code might scale with the magnitude of the current belief because there are just more trials when subjects are sure? Can you show this isn't the case by training it on balanced numbers of trials when subjects ought to be sure/unsure, perhaps? Or demonstrating that the numbers of such trials are already balanced?

To address this specific comment, we have first extracted the overall distributions of the current belief magnitude obtained through particle filtering – distributions that were used to bin the data on Fig. 7c (and Supplementary Fig. 11a). As can be seen below, the distributions of the current belief magnitude do not peak for high values of belief magnitude where participants can be expected to be certain, but rather for moderate values of belief magnitude (logLR values of 1.00-1.25 in both conditions) where participants are likely to be rather uncertain of the current value of the hidden state. This is precisely the level of cognitive uncertainty that we targeted to achieve with our experimental protocol, by using largely overlapping distributions for the generative stimulus distributions used for the two categories (depicted in Fig. 1a and replotted below):

Although we did not measure participants' confidence in our experimental protocol, these observations suggest that there were not two distinct classes of trials in the experiment ('sure' trials and 'unsure' trials), but rather a continuum of trials of varying levels of uncertainty regarding the current value of the hidden state. These smooth, unimodal distributions of current belief magnitude make it possible to study the relationship between the neural coding of consistency and the current belief magnitude reported in Fig. 7c.

To clarify the question regarding the number of trials used for training and testing the decoding algorithm, we have adopted for this particular analysis a 'common filters' approach, now described in the methods section. In practice, we used the same spatial MEG patterns (coding weights) for decoding consistency across all tested bins – each of which contained the same number of epochs (hence the variable distance between successive bins visible on Fig. 7c). These shared spatial patterns were estimated through cross-validation across all epochs/bins, separately for the cue-based and outcome-based conditions, such that the results obtained for the different bins could not be due to training on different numbers of trials where participants are sure vs. unsure:

“We accounted for the unbalance in trial number between sequences leading to response repetitions and response reversals when decoding consistency at each stimulus position (Fig. 6d). Because the smaller number of sequences leading to response reversals could artificially reduce the coding precision of consistency, we applied the same spatial MEG patterns for the two types of sequences ('common filters' approach). These spatial patterns (coding weights) were estimated through cross-validation based on sequences leading to response repetitions, separately for the cue-based and outcome-based conditions. For similar reasons, when assessing the effect of the current belief magnitude on the neural coding of stimulus characteristics (Fig. 7c), we applied the same spatial MEG patterns for each bin. These spatial patterns were estimated through cross-validation across all bins, separately for the cue-based and outcome-based conditions.”

In other words, and by construction, the results presented on Fig. 7c are already obtained by training the decoder “on balanced numbers of trials when subjects ought to be sure/unsure”, as prescribed by the reviewer. We hope that these additional details and clarifications in the revised manuscript support the validity of our approach for assessing the relation between the neural coding of consistency and the current belief magnitude.

Small points:

P3 - surely the variety of an apple is inferred from both sensory signals and long term memory - the apple's colour could be an example of pure sensory inference? Also is it worth mentioning that in the real world, most cue-based inference is also active (we actively sample information), so really this paper is about active vs passive inference rather than cue vs outcome inference?

Based on comments made by different reviewers, we have modified the first paragraph of the introduction section. We now focus much more explicitly on the difference in the degree of control over information sampling (p. 3) – the key difference between the cue-based and outcome-based conditions in our experimental protocol:

“Making accurate decisions in an uncertain environment requires inferring its properties from imperfect information^{1,2}. When categorizing an ambiguous stimulus, this ‘hidden-state inference’ process consists in identifying the generative cause of sensory cues. By contrast, when foraging rewards, inference concerns the expected outcomes of possible courses of action. A constitutive, yet rarely considered, difference between these two forms of inference lies in the degree of control over information sampling conferred to the decision-maker. Indeed, cue-based inference relies on the presentation of information to an observer interpreting a relevant property of its environment (here, the category of the presented stimulus), whereas outcome-based inference relies on the active sampling of information by an agent interacting with its environment to achieve a particular goal (here, maximizing rewards).”

We have defined ‘cue-based’ inference in this paragraph as being based on the “*presentation of information [cues] to an observer interpreting a relevant property of its environment*”. By contrast, we define ‘outcome-based’ inference as being based on the “*active sampling of information [outcomes] by an agent interacting with its environment to achieve a particular goal*”. We do agree with the reviewer that, outside the lab, perceptual decisions are equally likely as reward-guided decisions to involve active sampling. We now thus refrain from making an implicit mapping between cue-based (passive) inference and perceptual decision-making on one hand, and between outcome-based (active) inference and reward-guided decision-making on the other hand. Please note, however, that we also avoid referring to the cue-based condition as ‘passive’, since this condition is different from ‘forced-choice’ or ‘observational’ conditions where the observer is not making decisions based on the incoming evidence – something we stress in the discussion section (p. 20):

“Note that the inference process used in the cue-based condition is not equivalent to ‘observational’ learning – i.e., learning from or about another agent interacting with uncertain evidence^{19,20}. Indeed, participants were actively committing to interpretations regarding the source of evidence in both conditions, the sole difference being the presence or absence of instrumental control over the sampling of evidence.”

Together, we agree with the reviewer that it is important to focus the introduction section on the difference in instrumental control between cue-based and outcome-based inference that we can study using our experimental protocol.

| P40 - why were the MEG signals low pass filtered at 8 Hz? Doesn't this lose a lot of information?

MEG and EEG signals show a prominent peak in their spectrum around 10 Hz – the well-characterized alpha range. Power fluctuations in this frequency band are known to reflect the orientation of visual attention in visual cortex and the preparation of motor commands in the dorsal sensorimotor pathway, but the phase of alpha oscillations is much less locked to the ongoing sensory or cognitive events that we seek to study in our analyses of the MEG data. We thus focused our analyses to the lower part of the MEG spectrum that gives rises to most event-related responses – i.e., in the delta (1-4 Hz) and theta (4-8 Hz) bands – by removing higher-frequency content that is not expected to show phase locking to sensory or cognitive events. In practice, performing the decoding analyses on non-filtered MEG signals (i.e., keeping frequency content above 8 Hz) does not improve coding precision but rather adds small oscillations to the time course of neural coding precision in the alpha range (~10 Hz). This confirms that waveform information in the alpha range (i.e., phase information, not power information) does not contribute to the neural coding of the stimulus characteristics analyzed in the study. Besides, due to the 1/f nature of the power spectrum of MEG and EEG signals, removing frequency information above 8 Hz through low-pass filtering has only a limited effect on the studied signals (since high-frequency information has inherently smaller amplitude than low-frequency information).

| P44 - why was the prior mean over the hazard rate so high at 0.2? Do you get the same results if you use a lower or empirical prior?

We used relatively wide priors for all model parameters. We chose 0.2 as prior mean for the perceived hazard rate to cover a wide range of possible values between 0 (assuming a perfectly stable hidden state) and 0.5 (assuming purely random fluctuations of the hidden state across consecutive trials). These priors, which are of course shared across cue-based and outcome-based conditions, have little influence on the posterior distributions obtained at convergence of the particle MCMC procedure. Indeed, the reviewer is correct that reducing the mean of the prior for the perceived hazard rate to 0.125 (the true generative hazard rate of the task), or even changing the shape of the prior distribution to a gamma distribution (instead of a truncated normal distribution), has no notable impact on the results. In particular, the selective difference between perceived hazard rates in cue-based and outcome-based conditions is fully robust to these changes to its prior distribution.

REVIEWER COMMENTS

Reviewer #1 (Remarks to the Author):

The authors have done a great job in revising this paper. The theoretical background and implications, the added value of the MEG results (in pinpointing updating at the stimulus level), and the rationale for their focus on the consistency signal are all a lot clearer in the revised version. I also applaud their effort to perform a new control experiment ruling out the difference between prospective and retrospective inference as explanation for the main finding. This paper is now in excellent shape and I endorse publication.

Reviewer #2 (Remarks to the Author):

The authors have performed lengthy additional analyses and addressed my comments in a satisfactory fashion. I have a few remaining minor comments, which I believe can strengthen the exposition.

1) In the additional analyses performed (Supplementary Figure 4) it became apparent that participants are not at all sensitive to the actual hazard rate. It is unfortunate that the current design did not manipulate hazard rates within a larger range, so it is not obvious whether this lack of adaptation reflects coarse sensitivity or total lack of sensitivity to the hazard rate. Nevertheless, the current result raises doubts about framing the findings as reflecting a differential perception of the stability of the environment. Do participants care about the hazard rate in this task? While I understand that the authors phrase their findings within the context of a Bayesian inference model, and while I think that this phrasing is descriptively adequate, alternative mechanistic explanations should be discussed. Would for example a form of "confirmation bias" in the outcome-based condition (whereby stimuli from the opposite category are ignored) mimic the behavioural results?

2) It would be interesting to see how the perceived hazard rate evolves across blocks (the temporal order of blocks) in the two OB/CB conditions and whether the type of the very first block (OB/CB) influences these trajectories.

3) After reading the paper again I found that the differences between OC and CB do not become very clear in the main Text and in Figure 1. The actual instructions that participants received (described) in the Methods section) offer a much clearer description. I suggest to refer to the "decks" analogy in Figure 1 and in the Main Text.

Reviewer #3 (Remarks to the Author):

The authors have responded to all my concerns, in great detail. I have no more comments. Congratulations on a very nice paper.

Reviewer #2

The authors have performed lengthy additional analyses and addressed my comments in a satisfactory fashion. I have a few remaining minor comments, which I believe can strengthen the exposition.

We are happy that the reviewer found our additional analyses and responses to his/her earlier comments convincing. We hope to have addressed his/her new remaining minor comments below.

1) In the additional analyses performed (Supplementary Figure 4) it became apparent that participants are not at all sensitive to the actual hazard rate. It is unfortunate that the current design did not manipulate hazard rates within a larger range, so it is not obvious whether this lack of adaptation reflects coarse sensitivity or total lack of sensitivity to the hazard rate. Nevertheless, the current result raises doubts about framing the findings as reflecting a differential perception of the stability of the environment. Do participants care about the hazard rate in this task? While I understand that the authors phrase their findings within the context of a Bayesian inference model, and while I think that this phrasing is descriptively adequate, alternative mechanistic explanations should be discussed. Would for example a form of "confirmation bias" in the outcome-based condition (whereby stimuli from the opposite category are ignored) mimic the behavioural results?

As shown by our additional analyses, participants did not adapt their perceived hazard rate to the subtle, uncued change in hazard rate between blocks. The primary aim of this study was to compare hidden-state inference between the cue-based (Cb) and outcome-based (Ob) conditions, not to investigate the adaptation to varying hazard rates in either condition. Earlier work focusing on this question (Behrens et al., 2007, *Nat. Neurosci.*; Glaze et al., 2018; *Nat. Hum. Behav.*; Filipowicz et al., 2020, *eLife*; Piray and Daw, 2020, *PLOS Comp. Biol.*) has typically used much more drastic changes in hazard rate over time, which made these changes much more salient and explicit to participants.

Regarding the question as to whether participants care about the hazard rate in our task, we have run an additional experiment where the hazard rate is cued at the beginning of each individual block of trials (as "more frequent changes" or "less frequent changes"). If participants do not care about the hazard rate in this task, then pre-cueing the hazard rate should not affect the perceived hazard rate used by participants in the following block. By contrast, if participants care about the hazard rate, then they should show higher perceived hazard rates in blocks cued as more volatile ("more frequent changes") than blocks cued as more stable ("less frequent changes"). We have found that participants adjust their perceived hazard rates when cued about changes in hazard rate in the two conditions (both $p < 0.01$; see the additional figure on the next page). This finding confirms that participants do care about the hazard rate in this task, and that they are capable of adjusting their perceived hazard rate in the Cb and Ob conditions. Furthermore, and like the experiment reported in the manuscript, the difference in perceived hazard rate reported in our study remains significant for more stable and more volatile blocks when they are cued as such.

This additional experiment, conducted for a different purpose along with other datasets, indicates that participants care about the hazard rate to the same extent in both conditions when it is cued explicitly (they could have ignored these explicit cues about the hazard rate), but does not address the question of whether participants would be sensitive to larger

uncued changes in hazard rate. Note that this is not a claim that we make in the present manuscript. Nevertheless, we agree with the reviewer that it constitutes an interesting question for future research.

Additional Fig. 1 | Adaptation of perceived hazard rate to cued changes in hazard rate. In this additional experiment ($N = 31$ participants), the hazard rate is cued at the beginning of each individual block of trials – as “less frequent changes” (more stable blocks, left panel) or “more frequent changes” (more volatile blocks, right panel). Best-fitting perceived hazard rates adapt to cued changes in hazard rate in the Cb (blue bars) and Ob (purple bars) conditions. The perceived hazard rate is significantly lower in the Ob condition, for blocks cued as more stable (left panel) and blocks cued as more volatile (right panel). Bars and error bars indicate means \pm s.e.m.

We agree with the reviewer and appreciate the distinction between descriptive (normative) and mechanistic accounts of the observed difference between hidden-state inference in the Cb and Ob conditions. The Supplementary Fig. 3 added to the revised manuscript already uses a mechanistic account to rule out response repetition bias as an explanation for the difference between the two conditions. In this case, we showed that a larger fraction of the posterior belief at trial t propagates as prior belief at trial $t+1$ in the Ob condition (parameter α of Supplementary Fig. 2).

We also already discuss in the revised manuscript (bottom of page 20) the empirical evidence against a “confirmation bias” in the Ob condition – which we refer to as a “biased, choice-supportive filtering of evidence”. We agree with the reviewer that the term “confirmation bias” is clearer and more explicit, and we have modified the discussion accordingly (see below). In the discussion, we argue that “the strength of belief-inconsistent evidence could be decoded [from MEG signals] with equal precision across conditions”, something at odds with a confirmation bias which would decrease the weighting of belief-inconsistent evidence. This absence of difference between the neural coding of belief-consistent and belief-inconsistent evidence is also already illustrated in Supplementary Fig. 11b.

At the behavioral level, a confirmation bias in the Ob condition would be accompanied by a significant decrease in the sensitivity to evidence in the Ob condition, something which is absent from the human data (Fig. 2d). Nevertheless, to falsify more directly this alternative account of the difference between the two conditions, we have conducted an additional

analysis of the behavioral data. We fitted the differences in response reversal curves and response repetitions curves between the Cb and Ob conditions using two alternative accounts: 1. a change in perceived hazard rate in the Ob condition (as in Fig. 3), and 2. a confirmation bias in the Ob condition (controlled by a gain parameter γ assigned to belief-inconsistent evidence). As expected, a change in perceived hazard rate produced a minimal change in the sensitivity to evidence which matched the human data (interaction with participants: $F_{1,23} = 0.4, p = 0.539$). By contrast, a confirmation bias produced a more substantial decrease in the sensitivity to evidence in the Ob condition which did not match the human data (interaction with participants: $F_{1,23} = 10.0, p = 0.004$). This interaction ‘falsifies’ (in the sense described in Palminteri et al., 2017, *Trends Cogn. Sci.*) the most standard account of confirmation bias described by the reviewer. Nevertheless, we agree that future research should examine other mechanistic accounts of the difference between the Cb and Ob conditions.

We now mention explicitly the term “confirmation bias” in the discussion where this alternative account is being considered (bottom of page 20), and we mention the fact that the decrease in evidence sensitivity predicted by a confirmation bias in the Ob condition stands at odds with the matched evidence sensitivity shown by participants across conditions (pages 20–21):

“ The slower reversal learning in the outcome-based condition is also unlikely to arise from a biased, choice-supportive filtering of evidence described and reported across cognitive domains^{28–32}. Indeed, such ‘confirmation bias’ (which filters out belief-inconsistent evidence) predicts a decreased sensitivity to evidence in the outcome-based evidence – something which we did not observe. Another observation which stands at odds with a confirmation bias is that the strength of belief-inconsistent evidence could be decoded from MEG signals with equal precision across conditions. ”

We hope that the reviewer will find these slight modifications to the discussion useful, and thank him/her for prompting these clarifications.

2) It would be interesting to see how the perceived hazard rate evolves across blocks (the temporal order of blocks) in the two OB/CB conditions and whether the type of the very first block (OB/CB) influences these trajectories.

We performed the block-wise fits of the perceived hazard rate suggested by the reviewer. The task consists of four blocks of each condition (Cb, Ob), presented in the following order: A A B B A A B B, where half of the participants started the task with the Cb condition and the other half with the Ob condition. We ran a mixed-effects ANOVA with task condition {Cb, Ob} and task-wise block index {1, 2, 3, 4} as within-subject factors, and the condition of the very first block {Cb, Ob} as between-subject factor.

The perceived hazard rate showed a moderate decrease across blocks (main effect of task-wise block index: $F_{3,66} = 3.8, p = 0.014$; 1st block, Cb: 0.177, Ob: 0.138; 4th block, Cb: 0.139, Ob: 0.089). This effect did not interact with the robust decrease in perceived hazard rate in the Ob condition – the main behavioral finding reported in the manuscript (main effect of task: $F_{1,22} = 36.4, p < 0.001$; interaction with trial-wise block index: $F_{3,66} = 0.2, p = 0.880$). Furthermore, the condition of the very first block had no measurable effect on the perceived hazard rate ($F_{1,22} = 0.3, p = 0.605$), nor its trajectory (interaction with task-wise block index:

$F_{3,66} < 0.1$, $p = 0.995$). Finally, the condition of the very first block did not interact with the decrease in perceived hazard rate in the Ob condition (interaction with task: $F_{1,22} = 0.7$, $p = 0.782$).

These additional analyses confirm that the decrease in perceived hazard rate in the Ob condition is sustained over time during the task, and does not depend on the condition of the very first block. Given the absence of interaction between the moderate decrease in perceived hazard rate across blocks and our behavioral findings, we hope that the reviewer agrees to not adding this additional information to the manuscript, which does not improve on its central claims.

3) After reading the paper again I found that the differences between OC and CB do not become very clear in the main Text and in Figure 1. The actual instructions that participants received (described) in the Methods section) offer a much clearer description. I suggest to refer to the "decks" analogy in Figure 1 and in the Main Text.

We thank the reviewer for this suggestion, which we have implemented in the new version of the manuscript. We now refer to the "cards and decks" analogy in the caption of Fig. 1 and in the main text (page 5):

" In the cue-based (Cb) condition, participants were instructed to monitor the deck (category A or B) from which presented cards (oriented stimuli) are drawn (Fig. 1c). In the outcome-based (Ob) condition, the same participants were instructed to select the action (left or right key press) which draws cards from a target deck (counterbalanced across blocks). As indicated above, the hidden state (the drawn deck in the cue-based condition, or the target deck-drawing action in the outcome-based condition) reversed occasionally and unpredictably between trials – thereby requiring participants to adapt their behavior following each reversal. "

We hope that these final revisions to the manuscript have addressed the remaining minor comments of the reviewer.

REVIEWERS' COMMENTS

Reviewer #2 (Remarks to the Author):

All my comments were fully addressed. I therefore recommend acceptance. I would like to compliment the authors for their very hard work and insightful responses to my comments. This will be an impactful publication.

Reviewer #2 (Remarks to the Author):

All my comments were fully addressed. I therefore recommend acceptance. I would like to compliment the authors for their very hard work and insightful responses to my comments. This will be an impactful publication.

We thank the reviewer for his/her positive evaluation of our work, and for his/her insightful comments which have improved our manuscript.